# Genetic determinants of micronucleus formation in vivo

D. J. Adams[1✉], B. Barlas[2,3,22], R. E. McIntyre[1,22], I. Salguero[4], L. van der Weyden[1], A. Barros[1,4], J. R. Vicente[2,3], N. Karimpour[2,3], A. Haider[2,3], M. Ranzani[1], G. Turner[1], N. A. Thompson[1], V. Harle[1], R. Olvera-León[1], C. D. Robles-Espinoza[1,5], A. O. Speak[1], N. Geisler[1,4], W. J. Weninger[6], S. H. Geyer[6], J. Hewinson[1], N. A. Karp[1], The Sanger Mouse Genetics Project*, B. Fu[1], F. Yang[1], Z. Kozik[7], J. Choudhary[7], L. Yu[7], M. S. van Ruiten[8], B. D. Rowland[8], C. J. Lelliott[1], M. del Castillo Velasco-Herrera[1], R. Verstraten[1], L. Bruckner[9,10], A. G. Henssen[9,10,11,12], M. A. Rooimans[13,14], J. de Lange[13,14], T. J. Mohun[15], M. J. Arends[16], K. A. Kentistou[17], P. A. Coelho[18], Y. Zhao[2,3], H. Zecchini[19], J. R. B. Perry[17,19], S. P. Jackson[4,20] & G. Balmus[1,2,3,4,21✉]

Genomic instability arising from defective responses to DNA damage[1] or mitotic chromosomal imbalances[2] can lead to the sequestration of DNA in aberrant extranuclear structures called micronuclei (MN). Although MN are a hallmark of ageing and diseases associated with genomic instability, the catalogue of genetic players that regulate the generation of MN remains to be determined. Here we analyse 997 mouse mutant lines, revealing 145 genes whose loss significantly increases ($n = 71$) or decreases ($n = 74$) MN formation, including many genes whose orthologues are linked to human disease. We found that mice null for *Dscc1*, which showed the most significant increase in MN, also displayed a range of phenotypes characteristic of patients with cohesinopathy disorders. After validating the *DSCC1*-associated MN instability phenotype in human cells, we used genome-wide CRISPR–Cas9 screening to define synthetic lethal and synthetic rescue interactors. We found that the loss of *SIRT1* can rescue phenotypes associated with *DSCC1* loss in a manner paralleling restoration of protein acetylation of SMC3. Our study reveals factors involved in maintaining genomic stability and shows how this information can be used to identify mechanisms that are relevant to human disease biology[1].

Genomic instability with concomitant accumulation of extranuclear MN is a hallmark of many disorders including cancer[3], inflammatory-associated diseases[4,5] and ageing[6–8]. MN are chromosome fragments that are formed due to mitotic segregation errors[9] or unrepaired DNA breaks[10] leading to mitotic chromosome bridges and breakage–fusion–bridge events[11,12]. Protected by an atypical nuclear envelope[13], MN can exist for several cellular generations, acquire aberrant epigenetic chromatin marks that may persist for future cellular generations[14,15] and can replicate their DNA, albeit asynchronously and more slowly than nuclear DNA[16]. Furthermore, the MN nuclear envelope can rupture, leading to the accumulation of MN DNA damage and subsequent chromosomal recombination (chromothripsis)[2,17–19], as well as

a potent proinflammatory response through cGAS (cyclic GMP-AMP synthase)[4,5] which can result in cellular senescence[7,8]. Although the molecular mechanisms driving MN formation have been under deep scrutiny, knowledge of the genetic factors controlling MN formation in vivo is lacking.

## In vivo MN screen and human correlates

To identify factors that can regulate MN formation in vivo, we screened over 6,000 mice across 997 loss-of-function mutant lines, using a highly sensitive detection method that enumerates MN in red blood cells using flow cytometry[20] (Fig. 1a). Our analysis defined genes upon which

[1]Wellcome Sanger Institute, Cambridge, UK. [2]UK Dementia Research Institute at the University of Cambridge, University of Cambridge, Cambridge, UK. [3]Department of Clinical Neurosciences, University of Cambridge, Cambridge, UK. [4]The Gurdon Institute and Department of Biochemistry, University of Cambridge, Cambridge, UK. [5]Laboratorio Internacional de Investigación Sobre el Genoma Humano, Universidad Nacional Autónoma de México, Santiago de Querétaro, México. [6]Division of Anatomy, MIC, Medical University of Vienna, Wien, Austria. [7]Functional Proteomics Group, Chester Beatty Laboratories, The Institute of Cancer Research, London, UK. [8]Division of Cell Biology, The Netherlands Cancer Institute, Amsterdam, The Netherlands. [9]Experimental and Clinical Research Center (ECRC) of the MDC and Charité Berlin, Berlin, Germany. [10]Max-Delbrück-Centrum für Molekulare Medizin, Berlin, Germany. [11]Department of Pediatric Oncology and Hematology, Charité—Universitätsmedizin Berlin, corporate member of Freie Universität Berlin, Humboldt-Universität zu Berlin, Berlin, Germany. [12]German Cancer Consortium (DKTK), partner site Berlin, and German Cancer Research Center (DKFZ), Heidelberg, Germany. [13]Department of Human Genetics, Section of Oncogenetics, Amsterdam UMC, Vrije Universiteit Amsterdam, Amsterdam, The Netherlands. [14]Cancer Center Amsterdam, Cancer Biology and Immunology, Amsterdam, The Netherlands. [15]Division of Developmental Biology, MRC, National Institute for Medical Research, London, UK. [16]Division of Pathology, Cancer Research UK Scotland Centre, Institute of Genetics & Cancer The University of Edinburgh, Edinburgh, UK. [17]MRC Epidemiology Unit, Wellcome-MRC Institute of Metabolic Science, University of Cambridge School of Clinical Medicine, Cambridge, UK. [18]Department of Genetics, University of Cambridge, Cambridge, UK. [19]Metabolic Research Laboratory, Wellcome-MRC Institute of Metabolic Science, University of Cambridge School of Clinical Medicine, Cambridge, UK. [20]Cancer Research UK Cambridge Institute, Cambridge, UK. [21]Department of Molecular Neuroscience, Transylvanian Institute of Neuroscience, Cluj-Napoca, Romania. [22]These authors contributed equally: B. Barlas, R. E. McIntyre. *A list of authors and their affiliations appears at the end of the paper. ✉e-mail: da1@sanger.ac.uk; gb318@cam.ac.uk

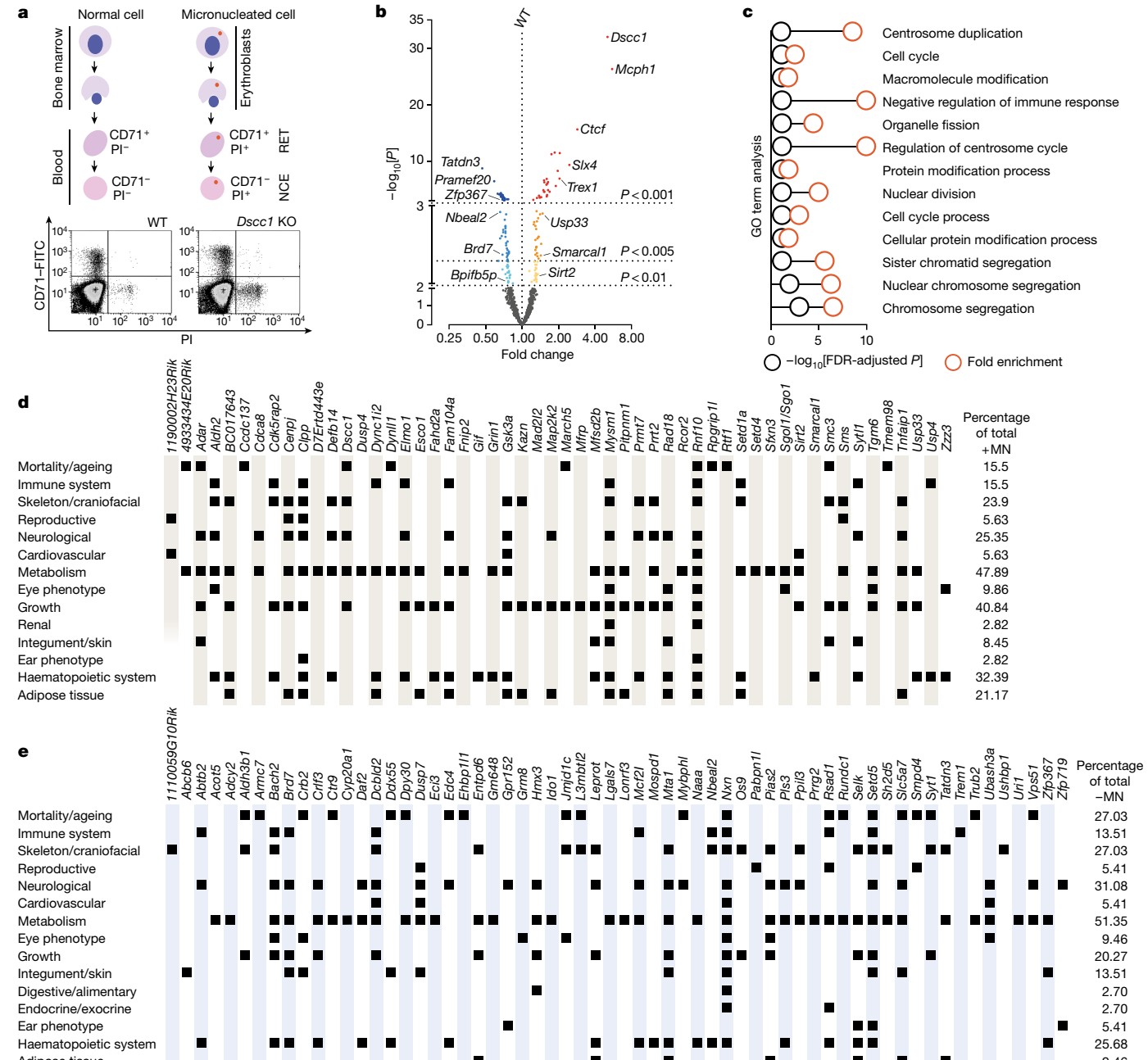

**Fig. 1 | An in vivo screen for genetic regulators of MN formation. a,** Schematic of the in vivo micronucleus assay. Full protocol details have been described previously[20]. Data for a *Dscc1*-KO mouse and a WT littermate control are shown. CD71, transferrin receptor; NCE, normochromatic erythrocyte; PI, propidium iodide; RET, reticulocyte. **b,** The MN screen results indicating mutants that, compared with the WT, have lower (−MN; left) or higher (+MN; right) MN formation and accumulation. Three statistical tiers are indicated on the basis of *P*-value cut-offs and false-discovery rates (FDR): tier 1 (*P* < 0.001; FDR < 0.017; +MN, red dots; −MN, dark blue dots); tier 2 (*P* < 0.005; FDR < 0.046; +MN, orange dots; −MN, blue dots); and tier 3 (*P* < 0.01; FDR < 0.068; +MN, yellow dots; −MN, light blue dots). The effect of genotype on the percentage of MN was assessed using a mixed linear effect beta regression model in R with baseline WT mice (*n* = 285) together with mice of each genotype. A total of *n* = 6,210 mice were analysed. Multiple testing was managed by adjusting the *P* values to

control the FDR (Methods). The full dataset and statistics are provided in Supplementary Table 1. **c,** Pathway analysis for +MN screen hits, aligning them with biological processes. GO, Gene Ontology. **d,e,** Statistically significant phenotypes of mouse lines with increased (+MN; **d**) or decreased (−MN; **e**) MN[59]. Out of 71 +MN mutant lines, 54 had additional phenotypes; out of 74 −MN mutant lines, 62 had additional phenotypes. The squares indicate the related organ system affected. The percentage representation of phenotypes within the +MN and −MN genes is shown on the right. The full dataset and statistical methods are available through the International Mouse Phenotyping Consortium (IMPC) (www.mousephenotype.org). The individual mouse was considered to be the experimental unit in these studies. The data presented are a snapshot from September 2023 (Methods). Tabular data are also available at GitHub (https://github.com/team113sanger/Large-scale-analysis-of-genes-that-regulate-micronucleus-formation/tree/main/Mouse_Phenotyping_Data).

disruption either increased (+MN) or decreased (−MN) MN formation and accumulation compared with wild-type (WT) control mice (Methods). Hits from the screen were separated into three tiers on the basis of their statistical significance: tier 1 (*P* < 0.001; 56 genes: 29 +MN and

27 −MN), tier 2 (*P* < 0.005; 49 genes: 23 +MN and 26 −MN) and tier 3 (*P* < 0.01; 40 genes: 19 +MN and 21 −MN) (Fig. 1b, Supplementary Table 1 and Supplementary Fig. 1). Importantly, *Mcph1*[21], *CenpJ*[22], *Slx4*[23] and *Trex1*[24], of which the human orthologues are known disease-associated

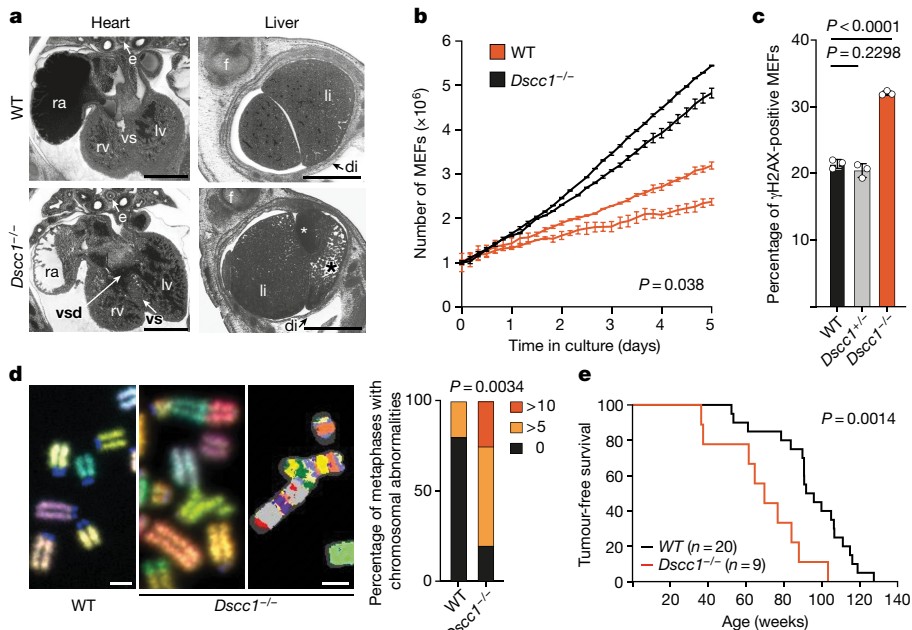

**Fig. 2 | Loss of *Dscc1* leads to early developmental defects and increased genomic instability. a**, Heart and liver abnormalities in *Dscc1⁻/⁻* E14.5 mouse embryos. The axial section (left; dorsal to the top) and sagittal re-sections (right; ventral to the top) were obtained through high-resolution episcopic microscopy (HREM) analysis of a *Dscc1*-mutant (bottom) and a WT (top) embryo. Bottom left, a ventricular septal defect (vsd) in a *Dscc1⁻/⁻* embryo. Bottom right, abnormal liver texture, specifically, a cyst (white asterisk) and abnormally enlarged liver sinusoids combined with a reduced number of hepatocytes (black asterisk) in the liver lobe of a *Dscc1⁻/⁻* embryo. di, diaphragm; e, oesophagus; li, liver; LV, left ventricle; RA, right atrial appendix; RV, right ventricle; VS, ventricle septum. Scale bars, 1 mm. Three embryos per genotype were analysed. **b**, Growth curves of primary mouse embryonic fibroblasts (MEFs) over 5 days in culture. Two independent WT and two independent *Dscc1⁻/⁻* MEF lines derived from littermate embryos are shown. $n = 3$ independent replicates each. Data are mean ± s.d. Statistical analysis was performed using

two-tailed Student's $t$-tests comparing the area under the curve (AUC) values. **c**, Flow cytometry analysis of MEFs, showing increased genomic instability, as measured by the presence of γH2AX-positive cells, an indicator of the presence of DNA damage. $n = 3$ biological replicates each. Data are mean ± s.d. Statistical analysis was performed using two-tailed Student's $t$-tests. **d**, Representative images of chromosomal abnormalities seen in primary MEFs of the indicated genotypes at passage 3 (left). Right, the percentage of abnormalities from chromosomal spreads comparing WT with *Dscc1⁻/⁻* MEFs. $n = 3$ biological replicates measuring $n = 10$ metaphases per genotype in each experiment. Statistical analysis was performed using two-way analysis of variance. Scale bars, 5 μm. **e**, Kaplan–Meyer analysis of *Dscc⁻/⁻* mice, showing that they have a decreased latency of tumour formation. Age and sex information are in provided in the Source Data. $n = 20$ (WT) and $n = 9$ (*Dscc1⁻/⁻*) mice. Statistical analysis was performed using log-rank (Mantel–Cox) tests.

genes, serve as positive controls with established roles in processes such as chromosome segregation, DNA damage response and chromothripsis. Most of the other hits have not been associated with MN formation (such as *Tnfaip1*) or are largely uncharacterized genes (for example, *Gm13125*), with many illuminating previously undescribed biology. This includes *Lsm10*, which was previously associated with snRNP processing[25] and is now linked to micronucleation. The −MN category is unique and has not been described previously. Mouse mutants/genes in this category proffer further investigation as processes such as aberrant erythropoiesis could explain mutants in this group (Extended Data Fig. 1a). To validate this category, we picked seven tier 1 −MN hit genes and used CRISPR–Cas9 editing to disrupt them in human CHP-212 cells, assessing MN formation using microscopy (Extended Data Fig. 1b). To examine reduced micronucleation, we exposed cells to a low chronic dose of hydroxyurea (HU; Methods), increasing the mean basal MN rate from 1.5% to 5.02% in WT (control) cells. In *DSCC1*-knockout (KO) and *TOP3A*-KO positive controls, the frequency of MN was 9.60% and 26.04%, respectively, while reduced micronucleation was observed after disrupting *TATDN3* (1.31%), *DUSP7* (2.19%) and *PIAS2* (2.91%), as expected (Extended Data Fig. 1b). These data highlight a rich collection of candidate genes in the −MN category that modify micronucleation in both mouse and human cells.

Analysis across multiple databases showed that our MN hits are part of a complex interconnected network (Fig. 1c and Extended Data Fig. 1c). To further understand the function of genes identified in our in vivo screen, we performed a comprehensive panel of phenotyping

analyses (consisting of over 200 unique parameters) of all of the mouse mutant lines, together with age- and sex-matched controls. This showed a key role for screen hits in maintaining homeostatic balance, with a substantial proportion of lines showing phenotypes associated with increased mortality as well as immune, metabolic, neuronal and skeletal dysfunction (Fig. 1d,e). To further assess the potential human relevance of hits from our MN screen, we integrated our findings with a genome-wide association study (GWAS) of mosaic loss of chromosome Y (LOY), a biomarker of defects in chromosome maintenance and the DNA damage response[26]. By mining over 200,000 healthy male human genomes from the UK Biobank[27], we found that LOY loci were enriched for the human orthologues of tier 1–3 MN genes ($P = 0.028$; MAGMA test) (Supplementary Table 2). Specifically, of the 137 tier 1–3 MN genes that mapped to human orthologues, 13 were LOY GWAS genes, while a further 6 were either expression or protein quantitative trait locus genes associated with LOY (Extended Data Fig. 2a and Supplementary Table 2). Importantly, a gene-based analysis (MAGMA[28]) showed that coding variants in *JMJD1C*, *SLX4*, *ENTPD6*, *EHBP1L1* and *NBEAL2* were aggregately and significantly associated with LOY (Supplementary Table 2). Furthermore, genes identified as +MN hits in our screen were orthologues of G2P/Deciphering Developmental Disorders[29] genes that were established to be disease causing for cancer, development and abnormalities of the eye and/or skin (Extended Data Fig. 2b and Supplementary Table 3). MN-associated genes were also found to be established contributors to tumorigenesis through somatic mutation (COSMIC tier 1[30]) or to be disease-associated in the GWAS catalogue,

and several have undergone de novo mutation in patients with developmental disorders[31] (Extended Data Figs. 1c and 2b). Collectively, these data highlight the relevance of many MN-associated genes to human disease phenotypes and traits. Through this unique resource, we provide genetic models that link MN formation in vivo to phenotypes and their sequela.

## *Dscc1* as a human disease model

One class of genes that we identified in the +MN group (Extended Data Figs. 1c and 3) comprised factors that are involved in sister chromatid cohesion (SCC) and included *Dscc1*, *Esco1*, *Smc3*, *Sgo1/Sgol1* and *Pds5b*[32]. Collectively, defects in the human counterparts of several of these genes cause multiorgan syndromes called cohesinopathies, including Cornelia de Lange syndrome (CdLS; *SMC3*) and chronic atrial and intestinal dysrhythmia (*SGOL1/SGO1*) that are associated with developmental and skeletal abnormalities, cardiovascular anomalies, visceral defects as well as behavioural and neurological disorders[33]. Notably, although *DSCC1* has not been reported to be causative of a cohesinopathy, it is approximately 3 Mb proximal to *RAD21*, another SCC gene, with both genes sometimes co-deleted in patients with CdLS[34]. We analysed data from around 500,000 individuals in the UK Biobank[27] (Methods) to examine the role of *DSCC1* in human phenotypes and disease. Focussed analyses identified common genetic variants associated with body mass index and bone mineral density (BMD), which appeared to confer their effects through altered *DSCC1* gene expression (Supplementary Table 4). Rare protein-truncating variants of *DSCC1*, independent of these common variants, also demonstrated a suggestive association with BMD (Supplementary Table 4). More broadly, additional common variant associations at the *DSCC1* locus were identified for adult height and vascular phenotypes (Supplementary Table 4).

As *Dscc1*-mutant mice displayed the most significant increase in MN (Fig. 1b) and because *DSCC1* is convincingly associated with human disease and traits, we chose to characterize this mouse mutant further. *Dscc1*-mutant mice were generated by targeted insertion of a gene-trap between exons 1 and 2 that results in transcript truncation and *Dscc1* disruption (hereafter, *Dscc1*[−/−] mice; *Dscc1*[tm1a(KOMP)Wtsi]) (Extended Data Fig. 4a–c). Notably, compared with littermate WT mice (WT; *Dscc1*[+/+]), *Dscc1*[−/−] mice were subviable most likely due to severe vascular anomalies of the heart and liver at embryonic day 14.5 (E14.5; Fig. 2a and Extended Data Fig. 4d–f). Surviving *Dscc1* mutant mice showed skeletal abnormalities, increased body weight, testicular atrophy with abnormal spermatogenesis that led to reduced fertility, increased bone mineral content as well as altered activity (Extended Data Fig. 5). Notably, these phenotypes relate to presentations seen in patients with cohesinopathies (Supplementary Table 5) and align with the above-mentioned analysis of UK Biobank data of human traits.

To seek an understanding of the cellular mechanisms that are responsible for MN formation and ensuing pathology, we isolated mouse embryonic fibroblasts (MEFs) from E13.5 embryos. Cultures of *Dscc1*[−/−] MEFs grew slower than those of WT littermate controls (Fig. 2b and Extended Data Fig. 6a–c) and had increased genomic instability, as measured by accumulation of the DNA-damage-response marker γH2AX (Fig. 2c). To test for structural chromosomal aberrations that could arise from this increased genomic instability, we performed multicolour fluorescence in situ hybridization (M-FISH) on chromosome spreads of *Dscc1*[−/−] MEFs and found extensive chromosome breakage and rearrangement events (Fig. 2d), some of which involved more than seven translocations/rearrangements within the same chromosome, reminiscent of chromothripsis. To further understand the consequences of such events in vivo, as chromosomal breaks can lead to rearrangements that promote cancer[3,11], we aged *Dscc1* mutant mice and WT controls. *Dscc1* mutants displayed significantly decreased tumour latency (P < 0.0014, log-rank test), with lymphoma being the

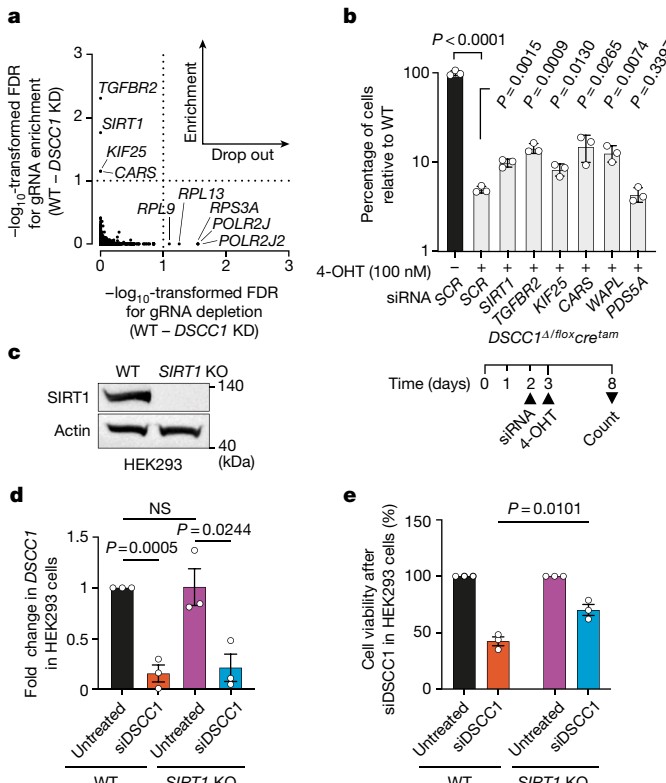

**Fig. 3 | Genetic rescue of cellular phenotypes associated with DSCC1 loss. a**, Classification of the most enriched/depleted CRISPR-target genes in *DSCC1*-mutant (KD) iPS cells as compared to isogenic WT controls. The dotted lines separate enriched and depleted hits and indicate the FDR thresholds. The raw data are available in the Source Data (the full analysis is available at GitHub). **b**, The effect of depleting the genes obtained from the *DSCC1*-KD CRISPR–Cas9 screen alongside the cohesin genes *WAPL* and *PDS5A*. RPE-1 *DSCC1* conditional KO cells (*DSCC1*[Δ/flox]*cre*[tam]) were transfected with either scrambled (SCR) siRNAs or siRNAs against the indicated gene in the presence of 100 nM 4-OHT; viability was assessed in comparison to the parental cell line (SCR; 4-OHT). The experiment was repeated *n* = 3 independent times (biological replicates in technical triplicate). The timeline of siRNA and 4-OHT addition is indicated. Note that the *y* axis is displayed on a log₁₀ scale. Data are mean ± s.d. Statistical analysis was performed using two-tailed Student's *t*-tests. **c**, Representative western blot analysis of SIRT1 expression in human WT and *SIRT1*-KO HEK293 cells. The experiment was repeated *n* = 3 independent times (biological replicates). **d**, siDSCC1 treatment of HEK293 cells leads to significantly reduced *DSCC1* transcript levels as measured using quantitative PCR with Taq-Man *DSCC1* probes (Methods). *n* = 3 independent experiments with *n* = 5 technical replicates each. Data are mean ± s.e.m. Statistical analysis was performed using two-tailed Student's *t*-tests; NS, not significant (*P* > 0.05). **e**, *SIRT1* KO rescues the siDSCC1 cell proliferation defect in HEK293 cells 3 days after *DSCC1* depletion. *n* = 3 biological replicates with *n* = 5 technical replicates each. Statistical analysis was performed using two-tailed Student's *t*-tests. Data are mean ± s.d.

predominant malignancy (Fig. 2e), therefore suggesting that *DSCC1* can act as a tumour suppressor.

Together with CHTF8 and CHTF18, DSCC1 is a component of the alternative replication factor C complex, RFC[CTF18], which loads PCNA (DNA-polymerase processivity clamp proliferating cell nuclear antigen) onto DNA during S phase of the cell cycle[35], bringing ESCO1 and ESCO2 acetyltransferases into close proximity of SMC3 to mediate SMC3 acetylation[36,37]. These processes are critical for replication fork processivity and the establishment of SCC up until anaphase[38]. To determine whether the loss of DSCC1 leads to MN formation in human cells, we generated *DSCC1*-mutant induced pluripotent stem cells (*DSCC1* knockdown (KD) iPS; Methods and Extended Data Fig. 7a,b).

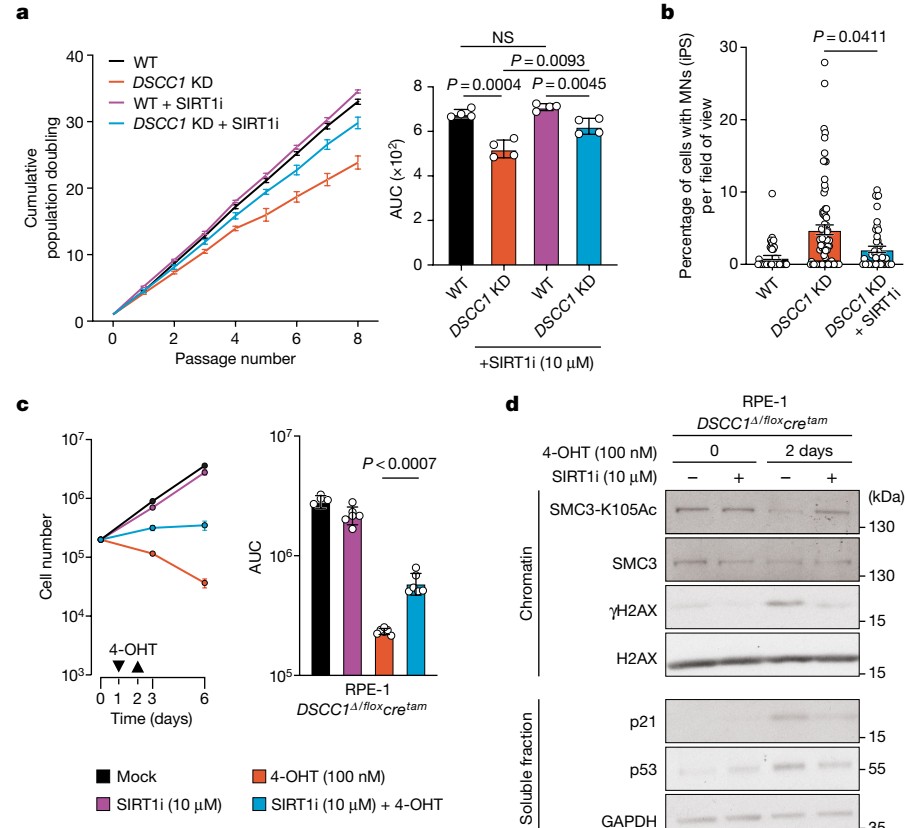

**Fig. 4 | SIRT1 inhibition rescues DSCC1-associated cellular phenotypes.**
**a**, SIRT1i rescues the proliferation defect of *DSCC1*-mutant cells and decreases MN formation and accumulation. The proliferation of human iPS cells in which *DSCC1* was disrupted (*DSCC1* KD) using CRISPR–Cas9 (Extended Data Fig. 7) was compared with control cells (WT; parental line) as well as cells treated with SIRT1i. Statistical analysis was performed using two-tailed Student's *t*-tests. *n* = 4 biological replicates. Data are mean ± s.e.m. **b**, SIRT1i (10 μM) treatment rescues MN formation in *DSCC1*-KD cells. Each dot represents an independent field of view. Data are mean ± s.e.m. Three biological replicates were performed. Significance was assessed by comparing the means of these experiments using a two-way Mann–Whitney *U*-test. **c**, Proliferation assay (left) and AUC (right) of

the RPE-1 *DSCC1^{Δ/flox}cre^{tam}* cell line in the presence of SIRT1i (10 μM) after *DSCC1* deletion by 4-OHT treatment (addition and removal indicated by arrows). Data are mean ± s.e.m. Statistical analysis was performed using two-tailed Student's *t*-tests, comparing the AUC for cells with and without SIRTi (10 μM) treatment. The experiment was performed three independent times (biological replicates) in duplicate. Significance was assessed by comparing the means of these experiments. **d**, Representative western blot images (from three independent/ biological replicate experiments) showing chromatin fractionation of the RPE-1 *DSCC1^{Δ/flox}cre^{tam}* cell line after the indicated treatments (uncropped images are shown in Supplementary Fig. 2).

As shown by mass spectrometry (MS), *DSCC1* disruption in these cells led to destabilization of the other two members of the RFC^{CTF18} complex and increased the abundance of proteins involved in the DNA damage response, such as H2AX and KDM6A, most likely as a response to the increased genomic instability (Extended Data Fig. 7c). As we saw in our mouse screen, loss of DSCC1 resulted in significantly increased MN formation, impaired SCC and a subsequent loss of fitness of human iPS cells (Extended Data Fig. 7c–f).

Together, these data show that, in mouse and human cells, DSCC1 is critical for genome maintenance and its deficiency leads to phenotypes that are associated with genomic instability.

## SIRT1 inhibition rescues *DSCC1* loss

To unbiasedly define genes and pathways that interact with DSCC1 loss, we performed a genome-wide CRISPR–Cas9 screen for genes that impact the proliferation of *DSCC1*-deficient human iPS cells. This analysis revealed four genes (*TGFBR2*, *SIRT1*, *KIF25* and *CARS*) that when disrupted could partially rescue the proliferation defect of *DSCC1* mutant iPS cells (phenotype suppressors), and five genes (*POLR2J2*, *POLR2J*, *RPS3A*, *RPL13* and *RPL9*) that when disrupted further decreased their proliferation/fitness (phenotype enhancers; drop-outs) (Fig. 3a). Of the phenotypic suppressors, *TGFBR2* is a putative tumour

suppressor gene that regulates the transcription of genes associated with cell proliferation[39], *SIRT1* encodes a NAD-dependent deacetylase that is known to deacetylate many proteins including histones (H1, H3 and H4), transcription factors (p53) and DNA repair proteins (Ku70 and PARP1)[40], *KIF25* encodes a member of the kinesin-like protein family required to prevent premature centrosome separation during interphase[41] and *CARS* encodes a cysteinyl-tRNA synthetase that ligates amino acid residues to their corresponding tRNAs for use in protein synthesis[42]. To validate the screen in an independent cell line, we first used hTERT-RPE-1 cells carrying a conditional *DSCC1* allele with *loxP* sites flanking exon 2 (*DSCC1^{Δ/flox}*)[43] into which we stably introduced a 4-hydroxytamoxifen (4-OHT)-inducible *cre* recombinase cassette, generating the *DSCC1^{Δ/flox}cre^{tam}* (*DSCC1* conditional KO) cell line (Extended Data Fig. 7g). *DSCC1* disruption in these cells resulted in severely impaired proliferation, with most cells becoming senescent or dead a few days after tamoxifen addition (Extended Data Fig. 7h). In this cell line, we next depleted the top suppressor genes from the CRISPR screen using short interfering RNAs (siRNAs), and found that depletion of *TGFBR2*, *SIRT1*, *KIF25* and *CARS* can partially rescue the lethality associated with DSCC1 loss (Fig. 3b). Notably, both WAPL and PDS5A bind to cohesin and have been shown to alleviate phenotypes associated with *DSCC1* loss[43], results concordant with our study (Fig. 3b and Extended Data Fig. 8a–c).

As SIRT1 is an attractive therapeutic target with clinic-ready inhibitors[44], we decided to focus our analysis on the relationship between DSCC1 and SIRT1. To validate our screen in a further model, we first used *SIRT1*-KO HEK293 cells (Fig. 3c) in parallel with *DSCC1* siRNAs (siDSCC1; Fig. 3d and Supplementary Fig. 2) and found that *SIRT1* KO partially rescues the cell viability defects induced by *DSCC1* depletion (Fig. 3e). Second, we used a potent and selective SIRT1 inhibitor (SIRT1i; EX 527, selisistat), which is more than 200-fold selective over SIRT2 and SIRT3 and has been shown to inhibit deacetylation of SIRT1 substrates both in cells and in vivo[45]. We first determined the dose of SIRT1i that fully inhibited SIRT1 by examining the levels of p53 Lys382 acetylation (p53-K382Ac), a bona fide SIRT1 substrate, after treating cells with ionizing irradiation[46] (Extended Data Fig. 9a). Importantly, incubation with SIRT1i at this dose did not affect $DSCC1^{\Delta/flox}cre^{tam}$-induced transcript depletion after addition of 4-OHT (Extended Data Fig. 9b). Notably, when *DSCC1*-KD iPS (Fig. 4a) or RPE-1 $DSCC1^{\Delta/flox}cre^{tam}$ cells were treated with SIRT1i, cellular proliferation was partially rescued and MN formation was reduced (Fig. 4a–c and Extended Data Fig. 9c), consistent with the above-mentioned experiments in *SIRT1*-KO HEK293 cells. Extending these analyses to other cohesinopathy-related genes, we used HU[47] to induce MN formation in the presence or absence of SIRT1i and in concert with cohesinopathy gene disruption. We found that SIRT1i can also partially alleviate MN formation in *RAD21*- and *SMC3*-KO lines (Methods and Extended Data Fig. 8d), but not in *HDAC8*- or *MAU2*-KO lines, a result consistent with the observation that *HDAC8* and *MAU2* mouse mutants did not have elevated levels of MN in our screen. Collectively, these data suggest that SIRT1 inhibitors should be investigated for potential repurposing in human cohesinopathies where micronucleation is a phenotype.

To gain further mechanistic insights, we next examined SMC3 acetylation at Lys105 on chromatin (SMC3-K105Ac), a process that was previously reported to be defective in $DSCC1^{\Delta/flox}$ cells[43]. Notably, SMC3 acetylation at Lys105 was restored after SIRT1i treatment (Fig. 4d; quantified in Extended Data Fig. 9d). SMC3 acetylation during S phase is believed to be critical for the stabilization of the SMC ring and consequent maintenance of SCC[36,37,43,48]. Consistent with our mouse data, deletion of *DSCC1* resulted in increased genomic instability as measured by γH2AX with SIRT1i restoring γH2AX to basal levels (Fig. 4d). To determine whether SIRT1 can directly deacetylate SMC3, we performed an in vitro assay using recombinant SIRT1 protein (rSIRT1). We first observed deacetylation of the known target Lys382 on p53 but, subsequently, we observed no effect of rSIRT1 on SMC3 in cells null for the SMC3 deacetylase HDAC8[49,50] (Extended Data Fig. 9e). Collectively, these results suggest that SIRT1 inhibition has an indirect effect on SMC3 acetylation. One SIRT1 target that could affect cellular survival and therefore shift the cohesin defect is p53 itself. In cells, p53 loss can rescue *DSCC1* KO essentiality and *DSCC1*/*TP53*-double-KO RPE-1 cells are viable in culture[51]. To determine whether p53 loss can rescue the cohesin defect seen in *DSCC1*-KO cells, and whether SIRT1i can influence this process, we quantified the percentage of metaphases containing railroad track and premature chromatin separation events. We show that RPE-1 *DSCC1*/*TP53*-double-KO cells retain a marked cohesin defect that is partially rescued by SIRT1 inhibition (Extended Data Fig. 9f), suggesting that the SIRT1 effect on cohesion is independent of p53. Finally, as SIRT1 is a deacetylase involved in many cellular functions that could influence the DSCC1–SMC3 pathway, including genome stability[52,53], transcriptional repression through histones[54], replication[55,56] and mitosis[57,58], we decided to perform acetylation analyses on chromatin fractions using mass spectrometry (MS). We first used *SIRT1-KO* and WT HEK293 cells with or without SIRT1i and confirmed the high specificity of the compound. We then analysed RPE-1 *TP53*-KO and RPE-1 *DSCC1*/*TP53*-double-KO cells in the presence or absence of SIRT1i (Supplementary Fig. 3) to reveal over 20 proteins that are selectively modified by SIRT1i in the *DSCC1*/*TP53*-double-KO background (Supplementary Fig. 4 and Supplementary Data). Notably, targets including SYMPK and SMARCA4 were rebalanced to WT acetylation levels after SIRT1i treatment (Supplementary Fig. 4 and Supplementary Discussion). These data highlight that SIRT1 operates at multiple levels, revealing new routes of investigation, not only for CdLS, but also for other cohesion-related degenerative disorders.

In conclusion, by screening almost 1,000 mouse mutants, we have identified more than 100 genes associated with MN formation, each representing a mouse model of genomic stability. These include $Dscc1^{-/-}$, a semi-viable mutant mouse with skeletal, neurological, reproductive and structural/developmental anomalies, as well as tumour predisposition. *DSCC1* disruption led to loss of cellular viability associated with dysregulated SMC3 acetylation that could be partially rescued by SIRT1 inactivation. Our data represent a resource of genetic determinants of genomic instability in vivo and provide a conceptual platform for the identification genetic and functional modifiers with relevance to human disease.

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

**The Sanger Mouse Genetics Project**

**Catherine L. Tudor¹, Angela L. Green¹, Cecilia Icoresi Mazzeo¹, Emma Siragher¹, Charlotte Lillistone¹, Diane Gleeson¹, Debarati Sethi¹, Tanya Bayzetinova¹, Jonathan Burvill¹, Bishoy Habib¹, Lauren Weavers¹, Ryea Maswood¹, Evelina Miklejewska¹, Michael Woods¹, Evelyn Grau¹, Stuart Newman¹, Caroline Sinclair¹, Ellen Brown¹, Brendan Doe¹, Antonella Galli¹, Ramiro Ramirez-Solis¹, Edward Ryder¹, Karen Steel¹, Allan Bradley¹, William C. Skarnes¹, David J. Adams¹, David Lafont¹, Valerie E. Vancollie¹, Robbie S. B. McLaren¹, Lena Hughes-Hallett¹, Christine Rowley¹, Emma Sanderson¹, Elizabeth Tuck¹, Monika Dabrowska¹, Mark Griffiths¹, David Gannon¹, Nicola Cockle¹, Andrea Kirton¹, Joanna Bottomley¹, Catherine Ingle¹, Chris Lelliott¹ & Jacqueline K. White¹**

**136** | Nature | Vol 627 | 7 March 2024

## Methods

### Animals

All experiments were performed in accordance with UK Home Office regulations and the UK Animals (Scientific Procedures) Act of 2013 under UK Home Office licences. These licences were approved by the Wellcome Sanger Institute (WSI) Animal Welfare and Ethical Review Board. Mice were maintained in a specific-pathogen-free unit under a 12 h light and 12 h dark cycle with lights off at 19:30 and no twilight period. The ambient temperature was $21 \pm 2$ °C, and the humidity was $55 \pm 10\%$. Mice were housed at 3–5 mice per cage (overall dimensions of caging: 365 mm × 207 mm × 140 mm (length × width × height), floor area, 530 cm²) in individually ventilated caging (Tecniplast, Sealsafe 1284L) receiving 60 air changes per hour. In addition to Aspen bedding substrate, standard environmental enrichment of Nestlets, a cardboard tube/tunnel and wooden chew blocks were provided. Mice were given water and diet ad libitum.

### Mouse generation

A complete list of the mouse lines used in this study is provided in the Source Data. Most mouse mutants were generated using the well-validated 'KO-first allele'. This strategy relies on the identification of an exon common to all transcript variants, upstream of which a LacZ cassette is inserted to make a constitutive KO/gene-trap known as a *tm1a* allele. In contrast to the *tm1a* allele, *tm1b* creates a frameshift mutation after Cre-mediated deletion of the *loxP*-flanked exon. Other allele types are also possible and have been described previously[60]. Mouse production was performed as described previously[61]. We maintained most mutant lines (73% of the mice tested in this study) on a pure inbred C57BL/6N background, with the other lines on mixed C57BL/6 backgrounds (for example, C57BL/6N;C57BL/6BrdTyrc-Brd). For the C57BL/6N background, a core colony was established using mice from Taconic Biosciences, which was refreshed at set generational points (typically ten generations) and cryopreserved at regular intervals to avoid genetic drift. The sex and age for all mice analysed is available in the Source Data. For tumour-watch studies, mice were aged for spontaneous tumour formation until they became moribund in keeping with the above-mentioned Home Office Guidelines. To ensure compliance, mice were examined twice daily for symptoms including weight loss, poor coat condition and hunched back. Tumour histology was analysed by a consultant pathologist to confirm cancer diagnoses. Mice were assigned randomly to groups on the basis of Mendelian inheritance.

### In vivo MN screen

The in vivo MN screen was performed according to a previously described protocol[20]. The samples were analysed on the LSRFortessa or Cytomics FC500 (Becton Dickinson) system with a minimum of 100,000 events collected per sample. The gating strategy used is shown in Supplementary Fig. 1. For the analysis of MN screening data, a mixed linear effect beta regression model exploring the effect of genotype on the percentage of MN, was used. This was implemented within R (glmmTMB, v.1.0.1). In detail, a regression model was fitted using flow. cytometer as a fixed effect to account for any differences arising from the instrumentation, while assay.date was fitted as a random effect to account for the variance introduced by batch ($Y$ - genotype + flow. cytometer + (1|batch)). The genotype effect and associated error were estimated as a marginal mean using the emmeans package (R; v.1.4.4). The significance of the genotype effect was assessed using a likelihood ratio test. Analysis code is available at GitHub.

### High-throughput phenotypic screen

The high-throughput phenotyping pipeline used was a series of standardized tests conducted in accordance with standard operating procedures (available at IMPReSS (https://www.mousephenotype.org/impress/index) and were performed by the Mouse Genetics Project (MGP) at the Wellcome Sanger Institute (WSI). Tests covered a broad range of biological areas, including metabolism, cardiovascular, neurological and behavioural, bone, sensory and haematological systems, and plasma chemistry. Factors predicted to affect the variables were standardized where possible. If this was not possible, measures were taken to reduce potential biases, for example, the impact of different people performing the test (known as the minimized operator), and the time of day of the test, as defined by the Mouse Experimental Design Ontology (MEDO)[62]. The data captured with the MEDO ontology can be found at http://www.mousephenotype.org/about-impc/arrive-guidelines. Moreover, pre-established reasons were defined for quality-control failures (for example, insufficient sample, error with equipment during test) and detailed using IMPRESS, and the data inclusion/exclusion criteria were therefore standardized. All discarded data were retained and tracked in a database to enable quality-control-failed data to be audited. Phenotyping data were collected at regular intervals on age-, sex- and strain-matched WT (control) mice. On average, at least seven homozygote mice of each sex per KO line were generated for phenotyping. If no homozygotes were obtained from ≥28 offspring from heterozygote intercrosses at postnatal day 14 (P14), the line was declared homozygous lethal. Similarly, if less than 13% of the pups resulting from heterozygote intercrosses were homozygous at P14, the line was judged as being homozygous subviable. In this event, heterozygote mice were examined in the phenotyping screen. The random allocation of mice to experimental group (WT versus KO) was driven by Mendelian inheritance. Owing to the high-throughput nature of the phenotyping screen, blinding the operators to the identity of KO lines during phenotyping was not used as the cage cards used to identify the mice included genotype information. However, in a high-throughput environment without a defined hypothesis, the potential bias is minimized. In all cases, the individual mouse was considered to be the experimental unit. Further experimental design strategies (for example, exact definition of a control animal) are defined using a standardized ontology as described previously[62] and are available from the IMPC portal (http://www.mousephenotype.org/about-impc/arrive-guidelines). For a few lines, phenotyping data were also generated on a mutant of the same gene at another IMPC phenotyping centre and used to augment/enrich phenotypes from WSI. In figures that show phenotyping data, if the same phenotype was assessed by multiple assays, the most statistically robust result is shown.

### Characterization of MN gene candidates in human datasets

MN gene candidates were mapped to orthologous genes in the human genome using ENSEMBL and integrated with GWAS data on mosaic LOY[26]. This was performed using PAR-LOYq calls from 205,011 male participants from the UK Biobank study[27]. An enrichment analysis was performed across the whole dataset to test for the over-representation of MN genes at LOY GWAS loci. To do this, we first performed MAGMA analyses (v.1.08)[28] using all genomic variants within each MN gene extracting gene-level associations to the LOY phenotype. Genes were annotated on the basis of their proximity to genome-wide significant loci ($P < 5 \times 10^{-8}$) associated with LOY, specifically if they were 500 kb up- or downstream of the LOY gene start or end position. Second, further MAGMA analyses were performed using only those variants that were predicted to have deleterious effects (for example, non-synonymous and loss of function). Genes exhibiting an FDR-corrected MAGMA $P < 0.05$ were considered to be significant. Finally, for genomic loci reaching at least a suggestive level of significance in the GWAS ($P < 5 \times 10^{-5}$), we performed SMR and HEIDI tests (v.1.02)[63] using blood gene expression level data from the eQTLGen study[64] and blood protein level data from the Fenland study[65]. For both datasets, we considered expression of a gene to be influenced by the same genomic variation as that seen in the LOY GWAS if the FDR-corrected $P$ value for the SMR test was $P < 0.05$ and the $P$ value for the HEIDI test was $P > 0.01$. Human genomic variation within or around the *DSCC1* gene was further studied

by querying associations towards the human-equivalent phenotypic traits to those observed in *Dscc1*-mutant mice. Specifically, GWAS on BMD[66], body mass index, number of children ever born[67] and LOY[26] were used to ascertain gene-level associations using all available variants within the *DSCC1* gene and to perform SMR and HEIDI tests against the eQTLGen data, as described above (Supplementary Table 4). For the same four traits, exome gene-burden tests were performed using phenotypic and genetic data from the UK Biobank study. Rare exome variants (minor allele frequency < 0.1%) were identified on the basis of their predicted consequence on protein function and, using VEP[68] and LOFTEE[69], high-confidence protein truncation variants within *DSCC1* were collapsed and tested for associations towards the four traits using BOLT-LMM[70,71] (Supplementary Table 4). Finally, a phenome-wide association study for common variants within *DSCC1* was performed using the Open Targets Genetics Portal[72] (Supplementary Table 4).

## HREM analysis

For analysis with HREM, embryos were collected at E14.5 and fixed in Bouin's solution overnight. After washing in PBS, the embryos were dehydrated in a graded series of methanol. They were then infiltrated and embedded in methacrylate resin (JB4, Polysciences Europe) and stained with eosin B and acridine orange, according to previously published protocols[73]. The polymerized resin blocks were analysed using HREM resulting in volume datasets with isotropic voxel sizes of 2.55–3 µm. Visualization and further analysis of the HREM data were performed using Amira v.6.7.0 (Thermo Fisher Scientific) and OsiriX (v.5.6, 64 bit, Pixmeo). The embryos were staged and systematically screened for abnormalities according to a standardized protocol[74,75].

## Cell lines

MEFs were prepared from E13.5 embryos, after timed matings between *Dscc1*[+/-] mice. In brief, embryos were dissected from the decidua, mechanically disrupted and cultured in DMEM supplemented with 10% fetal bovine serum (FBS), 1.0 mM L-glutamine, 0.1 mM minimal essential medium, non-essential amino acids and penicillin–streptomycin. The initial plating was defined as passage zero, and cells were subsequently maintained on a standard protocol[76]. *SIRT1*-KO HEK293 cells were obtained from Kerafast (ENH131-FP). Cells were grown in DMEM supplemented with 10% FBS, penicillin–streptomycin and 1% GlutaMAX. RPE-1 *DSCC1*[Δ/flox] cells were obtained from Jallepalli Laboratory[43], and RPE-1 *TP53*-KO and *TP53/DSCC1*-double-KO cells were obtained from the de Lange laboratory[51]; these cell lines were grown in DMEM supplemented with 10% FBS and 1% GlutaMAX. iPS cells were grown in Tesr-E8 supplemented with 10 µM Y-27632 ROCK inhibitor (Stem Cell). HAP1 cells[77] were cultured in Iscove's modified Dulbecco's medium (Invitrogen), supplemented with 10% FCS (Clontech), 1% Ultra-Glutamin (Lonza) and 1% penicillin–streptomycin (Invitrogen). *ΔPDS5A* and *ΔWAPL* HAP1 cells were generated using CRISPR–Cas9 as described previously[78,79]. The CHP-212 neuroblastoma cell line (CRL-2273) was grown in RPMI with 10% FBS.

Modification of human iPS cells was performed according to established protocols[80]. In brief, the Gene Editing facility at WSI generated the *DSCC1*-KD BOB/iPS lines. We believe these cells to be null with just 2–3% of protein expression retained (Extended Data Fig. 7) but, nonetheless, designate this a KD allele. An asymmetrical exon within the target gene was replaced with a puromycin cassette, and a frameshift indel was introduced into the other allele. A template vector containing an EF1a-puromycin cassette was constructed for each gene, incorporating two 1.5 kb homology arms designed to align with the sequence surrounding the targeted exon. Two guide RNAs (gRNAs) were designed for each exon (Extended Data Fig. 7). The template vector (2 µg), both gRNA vectors (3 µg) and hSpCas9 (4 µg) were transfected into $2 \times 10^6$ cells using the Human Stem Cell Nucleofector Kit 2 (VPH-5022, Lonza). Subsequently, cells were seeded in 10 cm² dishes and, after 72 h, they underwent selection with 3 µg ml⁻¹ puromycin. Single cells were

then expanded and subjected to genotyping for the verification of a frameshift indel using Sanger sequencing. The resulting KO lines were cultured in the presence of 1 µg ml⁻¹ puromycin (ant-pr-1, InvivoGen). All of the cell lines in the research laboratories that participated in this study are routinely tested for mycoplasma and STR profiled and/or validated on the basis of the presence of unique engineered alleles as described in the Reporting Summary.

## Chromosome preparation and FISH

Metaphase preparations were performed using a standard protocol[81]. For M-FISH analysis, mouse-chromosome-specific DNA libraries were provided by the Flow Cytometry Core Facility of Wellcome Sanger Institute[82]. To make 10 tests of M-FISH probe, 500 µl of sonicated DNA was precipitated with 100 µl mouse Cot-1 DNA (Invitrogen) and resuspended in 120 µl hybridization buffer (50% formamide, 2× saline-sodium citrate (SSC), 10% dextran sulfate, 0.5 M phosphate buffer, 1× Denhardt's solution, pH 7.4). Metaphase preparations were dropped onto precleaned microscopy slides, and then fixed in acetone (Sigma-Aldrich) for 10 min followed by dehydration through an ethanol series (70%, 90% and 100%). Metaphase spreads on slides were denatured by immersion in an alkaline denaturation solution (0.5 M NaOH, 1.0 M NaCl) for approximately 40 s, followed by rinsing in 1 M Tris-HCl (pH 7.4) solution for 3 min, 1× PBS for 3 min and dehydration through a 70%, 90% and 100% ethanol series. The M-FISH probe (10 µl for each 22 × 22 mm hybridization area) was denatured at 65 °C for 10 min before being applied onto the denatured slides. The hybridization area was sealed with a 22 × 22 mm² coverslip and rubber cement. Hybridization was performed in a 37 °C incubator for 40–44 h. The post-hybridization washes included a 5 min stringent wash in 0.5× SSC at 75 °C, followed by a 5 min rinse in 2× SSC containing 0.05% Tween-20 (VWR) and a 2 min rinse in 1× PBS, both at room temperature. Finally, the slides were mounted with SlowFade Diamond Antifade Mountant containing 4′,6-diamidino-2-phenylindole (DAPI, Invitrogen). Images were visualized on the Zeiss AxioImager D1 fluorescence microscope equipped with narrow band-pass filters for DAPI, DEAC, FITC, CY3, TEXAS RED and CY5 fluorescence and an ORCA-EA CCD camera (Hamamatsu). M-FISH digital images were captured using the SmartCapture software (Digital Scientific UK) and processed using the SmartType Karyotyper software (Digital Scientific). At least 20 metaphases from each sample were fully karyotyped on the basis of M-FISH and enhanced DAPI banding.

## CRISPR–Cas9 screen and sequencing

WT and *DSCC1*-KD iPS cells ($1 \times 10^8$) were independently infected with a human genome-wide guide RNA (gRNA) lentiviral library[83] that had been recloned to swap the puromycin-resistance cassette with a neomycin-resistance cassette. Both lines were infected at a multiplicity of infection of 0.1–0.2 and a library coverage of 500× in three independent replicates, which were kept independent throughout the screen. Three days after infection, 1 mg ml⁻¹ G418 was added to the medium and cells were cultured for an additional 10 days. When cells required passaging, a minimum of $5 \times 10^7$ cells per technical replicate was maintained at a library coverage of 500×. From each replicate, PCR was performed to amplify the gRNA region, and gRNAs were sequenced as described previously[83]. Single-end Illumina sequencing reads of 19 nucleotides were counted for each gRNA. To identify depleted and enriched genes in the *DSCC1*-KD iPS cells the software package MAGeCK[84] v.0.5.6 was used. Extensive quality control of the screen was performed, and this analysis is available at the GitHub for this project (https://github.com/team113sanger/ Large-scale-analysis-of-genes-that-regulate-micronucleus-formation/ tree/main/CRISPR_screen_QC).

**Mini-arrayed CRISPR analyses.** The CHP-212 cell line was transduced with the lentiviral Cas9 plasmid (Addgene, 52962) and selected with

5 µg ml$^{-1}$ blasticidin (Thermo Fisher Scientific, 61120) for 5 days. To test the expression and cutting efficiency of Cas9, we took transformed and untransformed cells and further transduced them with a lentiviral BFP-GFP reporter virus (Addgene 67980). After 4 days, the cells were analysed using flow cytometry (CytoFLEX, Beckman Coulter) and the cutting and transduction efficiency were determined on the basis of the ratio of BFP- and GFP-positive cells as previously described[85]. Notably, we confirmed that cells continued to cycle and grow throughout the experiment.

The sgRNA-BFP plasmids were from the arrayed Sanger Institute CRISPR library (Sigma-Aldrich, HSANGERV; the sequences are provided in Supplementary Table 6). Bacteria were grown in 5 ml of LB medium overnight and DNA was extracted using a DNA purification kit (Amresco) and AcroPrep Advance 96-well filter plates (Cytavia, 8032). DNA concentrations were measured using the Quant-iT Pico-Green dsDNA Reagent (10535213). For each gene, two DNA vectors containing unique sgRNAs were mixed at equal amounts and then diluted to the same concentrations and blinded. Virus was produced by transfection of the mix of sgRNAs and the packaging plasmids psPAX (Addgene, 12260) and pMD2.G (Addgene, 12259) into HEK293FT cells. Virus was collected 3 days after transfection, and the viral titre was determined by measuring BFP expression using flow cytometry (CytoFLEX, Beckman Coulter). For the arrayed targeting screen, cells were seeded into PhenoPlate 96-well plates (Perkin Elmer, 6055302), leaving the outer wells blank. After the cells had adhered, they were transduced with lentivirus at a multiplicity of infection of >80% (each gene is targeted by two distinct gRNAs to increase the KO efficiency to >80% in our hands). Cells were allowed to recover before the addition of 2 µg ml$^{-1}$ of puromycin (Santa Cruz Biotechnology, sc-108071) for 48 h. After recovery, CHP-212 cells were treated with 12.5 µM of hydroxyurea (Merck, H8627) for three cell doublings and hTERT RPE-1 cells with 50 µM for 72 h (Supplementary Table 7 (HU titration)). Next, cells were fixed with 4% PFA (Alfa Aesar, 43368) and stained with TOPRO3 (Thermo Fisher Scientific, T3605). Cells were imaged using the Operetta CLS system (Perkin-Elmer) and analysed using the Harmony software (Imaging facility, CRUK Cambridge). A prescan (×5 air objective) of each well was performed to determine 180 fields of view of each well with the ideal seeding density. These fields of view were then reimaged using a ×40 water objective. For the analysis, z planes were transformed into a maximum projection, a sliding paraboloid filter was used, and the find nuclei and find cytoplasm functions were optimized to detect our cell lines in culture. Furthermore, the find spots function was used to find MN located in the cytoplasm. Other particles were excluded on the basis of staining intensity, roundness and size. Deblinding was performed after statistical analyses.

### *SIRT1* KO rescue cell viability experiment
HEK293 cells were grown in antibiotic free medium for two passages before seeding at a density of $1 \times 10^4$ cells per well in quintuplicate into 96-well plates. Then, 24 h later, ON-TARGETplus Human *DSCC1* (79095) siRNA-SMARTpool (L-014300-00-0005) at 25 nM final concentration was added to the cells, along with 0.2 ml per well of Dharmafect 2 (Dharmacon) transfection reagent in serum-free medium. The next day, complete antibiotic-free medium was added. Then, 48 h after transfection, the medium was refreshed on all wells with complete antibiotic-free medium. Three days after transfection, the cell viability was determined using the Promega Cell Titer Glo 2.0 cell viability assay. Medium was aspirated from wells and 175 µl of medium along with 25 µl Cell-Titer Glo reagent were added to each well and left to incubate for 10 min at room temperature. Medium and cell viability reagent mixture (150 µl) was transferred to black-walled, clear and flat-bottom 96-well plates for reading. Luminescence was read on the CLARIOstar microplate reader (BMG LABTECH). Cell viability was calculated by normalizing to untransfected control wells.

### *DSCC1* transcript analyses
RNA extraction was performed using the Monarch total RNA miniprep kit (New England BioLabs). RNA was converted to cDNA using the High-Capacity RNA-to-cDNA kit (Thermo Fisher Scientific) using 500 ng of total RNA. Gene expression was measured on the Quant-Studio 5 qPCR System (Thermo Fisher Scientific) using TaqMan gene expression assays for human *DSCC1* (Hs00900361_m1) or mouse *Dscc1* (Mm01195386_m1). TaqMan Universal Master Mix II with UNG-1 was used (Thermo Fisher Scientific; 4440038). Amplification parameters were as follows: 50 °C for 2 min; 95 °C for 10 min; followed by 40 cycles of 95 °C for 15 s and 60 °C for 60 s. Relative gene expression was determined on the basis of the $\Delta C_t$ values between the gene of interest and housekeeping genes *GAPDH* (Hs02786624_g1) and 18S rRNA (Hs03003631_g1) using the Design & Analysis v.2.6.0 software from Applied Biosystems (Thermo Fisher Scientific).

### Antibodies
The following antibodies were used: anti-CD71-FITC (SouthernBiotech, 1720-02, 0.5 mg ml$^{-1}$, 1:500)[20], anti-SIRT1 (rabbit, Cell Signalling, 2496S, 1:1,000), anti-centromere (Antibodies, 15-234-0001, 1:1,000), anti-rabbit Alexa 488 (Thermo Fisher Scientific, A11034, 1:2,000), goat anti-human Alexa 647 (Thermo Fisher Scientific, A21445, 1:2,000), anti-DSCC1 (H0079075-B01P, Novus Biologicals, 1:1,000), anti-HSP90 (F-8, Santa Cruz, 1:10,000), anti-HP1γ (05-690, Millipore, 1:1,000), goat-anti-mouse-PO (DAKO, P044701, 1:2,000), anti-SMC3 (Abcam, AB 9263, 1:250), anti-SMC3 (Thermo Fisher Scientific, A300-060A, 1:1,000), anti-acetyl SMC3 mouse (Sigma-Aldrich, MABE1073, 21A7, Lys105/106, 385016, 1:1,000), anti-p53 (Cell Signaling Technology, 1C12, 2524S, 1:1,000), anti-acetyl p53 (p53-K382Ac, Abcam, ab75754, EPR358(2) to p53 acetyl K382, 1:1,000), anti-phosphorylated-histone H2A.X (Ser139) (JBW301, Sigma-Aldrich, 05-636-I, 1:1,000), anti-β-actin (Merck, A5441, 1:10,000, 5% milk), anti-GAPDH (6C5, Abcam, ab8245, 1:1,000), anti-p21 (Abcam, ab109520, 1:1,000). Uncropped western blots are provided in the Supplementary Information.

### Immunoprecipitation
Flash-frozen cell pellets were thawed on ice and resuspended in 1 ml cell lysis buffer (25 mM Tris-HCl pH 7.4, 150 mM NaCl, 1 mM EDTA, 1% NP-40, 5% glycerol) freshly supplemented with 1:100 Pierce Universal Nuclease (Thermo Fisher Scientific, 88702), 1 mM DTT (Thermo Fisher Scientific, A39255) and Halt protease (Thermo Fisher Scientific, 1860932) and phosphatase inhibitor (Roche, PhosSTOP, REF: 04906845001) and incubated on ice for 30 min. The lysis buffer was also used as wash buffer. Protein was collected by centrifugation (15,000 rcf, 10 min at 4 °C), the supernatant was transferred to a fresh tube and the pellet was discarded. The protein concentration was measured using the Pierce BCA Protein Assay Kit (Thermo Fisher Scientific, 23225) according to the manufacturer's protocol. To start the immunoprecipitation, beads (Thermo Fisher Scientific, Immunoprecipitation Kit Dynabeads Protein A, 10006D) were conjugated to the antibody according to the manufacturers protocol. After optimization, 50 µl of beads were used to conjugate 2 µg of total antibody. The protein sample was diluted (using the lysis buffer) to 1 mg ml$^{-1}$ for immunoprecipitation and 1 ml of this sample was added to 50 µl of antibody conjugated beads. The protein–bead–antibody mixture was incubated on a rotator overnight at 4 °C. The sample was placed onto a magnet and the supernatant was transferred to a new tube (this was the flow-through that was retained to assess the antibody–bead uptake). The sample was washed on a rotator three times for 10 min in 1 ml lysis buffer at room temperature. In between the second and third wash, the sample was moved to a new Eppendorf tube to eliminate any proteins stuck to the tube. A single PBS wash was performed to the sample for 5 min on a rotator at room temperature, then the

sample was placed onto the magnet for the supernatant to be removed. The result was assessed using western blotting. To prepare the reagents for this, 50 µl of 2× SDS loading buffer and 5 µl of 10× reducing buffer were added to the beads. The input and the flow-through were prepared by adding the correct amount of protein, 4× SDS loading buffer, 10× reducing buffer and lysis buffer to volume. These samples were boiled at 95 °C for 5 min then loaded onto the gel (Bio-Rad, 4–12% gel) and run at 180 V for 45 min.

## Full proteome analysis

The samples were lysed in RIPA buffer plus HaltTM protease and phosphatase inhibitor cocktail (final concentration 2×, ThermoFisher Scientific) with probe sonication and heating. Samples were then centrifuged at 13,000 rpm for 15 min to remove the pellet. Protein concentrations were measured using a Pierce BCA protein assay (Thermo Fisher Scientific). A total of 100 µg of protein per sample was taken. Proteins were reduced by addition of TCEP (Tris(2-carboxyethyl) phosphine, Sigma-Aldrich), alkylated by iodoacetamide (Sigma-Aldrich) and then purified by trichloroacetic acid precipitation. Purified proteins were digested in 100 mM TEAB by trypsin (Thermo Fisher Scientific) at 1:25 (by weight) at 37 °C for 18 h. A total of 40 or 50 µg of peptides were labelled using 0.4 mg TMT10plex (Thermo Fisher Scientific) according to the manufacturer's instructions. The samples were mixed, dried in a SpeedVac and then fractionated on the XBridge BEH C18 column (2.1 mm inner diameter (i.d.) × 150 mm, Waters) with a gradient of 5% acetonitrile/0.1% $NH_4OH$ (pH 10) to 35% $CH_3CN$/0.1% $NH_4OH$ in 30 min (total cycle 60 min). The flow rate was at 200 µl min$^{-1}$. The peptides were reconstituted in 0.1% formic acid/$H_2O$ and analysed on the Orbitrap Fusion hybrid mass spectrometer coupled with the Ultimate 3000 RSLCnano system (both from Thermo Fisher Scientific). The samples were first loaded and desalted onto a PepMap C18 nano trap (100 µm i.d. × 20 mm, 100 Å, 5 µm; Thermo Fisher Scientific), then peptides were separated on the PepMap C18 column (75 µm i.d. × 500 mm, 2 µm; Thermo Fisher Scientific) over a linear gradient of 4–33.6% $CH_3CN$/0.1% formic acid in 180 min, with a cycle time of 210 min and a flow rate at 300 nl min$^{-1}$. The MS acquisition used MS3-level quantification with Synchronous Precursor Selection (SPS) with the top speed 3 s cycle time. In brief, the Orbitrap full MS survey scan was $m/z$ 380–1,500 with a resolution of 120,000 at $m/z$ 200, with AGC set at $4 × 10^5$ and 50 ms maximum injection time. Multiply charged ions ($z = 2–6$) with an intensity threshold at 5,000 were fragmented in an ion trap at 35% collision energy, with AGC set at $1 × 10^4$ and 50 ms maximum injection time, and isolation width of 0.7 Da in quadrupole. The top ten MS2 fragment ions were SPS selected with an isolation width of 0.7 Da, and fragmented in higher-energy collisionally activated dissociation (HCD) at 60% normalized collision energy (NCE), and detected in the Orbitrap to obtain reporter ion intensities at a better accuracy. The resolution was set at 60,000, and the AGC set at $6 × 10^4$ with maximum injection time at 105 ms. The dynamic exclusion was set 60 s with a ±7 ppm exclusion window. The raw files were processed using Proteome Discoverer v.2.4 (Thermo Fisher Scientific) using the Sequest HT search engine. Spectra were searched against fasta files of reviewed UniProt *Homo sapiens* entries (December 2021) and an in-house contamination database. The search parameters were as follows: trypsin with 2 maximum miss-cleavage sites; mass tolerances at 30 ppm for precursor and 0.6 Da for fragment ions; dynamic modifications of deamidated (N, Q) and oxidation (M); static modifications of carbamidomethyl (C) and TMT6plex (peptide N-terminus and K). Peptides were validated by Percolator with the $q$ value set at 0.01 (strict) and 0.05 (relaxed). The TMT10plex reporter ion quantifier included 20 ppm integration tolerance on the most confident centroid peak at the MS3 level. Only unique peptides were used for quantification. The co-isolation threshold was set at 100%. Peptides with average reported $S/N > 3$ were used for protein quantification, and the SPS mass matches threshold was set at 50%.

## Chromatin enrichment and MS analysis

Flash-frozen cell pellets were thawed on ice and resuspended in nuclear-extraction buffer (15 mM Tris-HCl pH 7.5, 60 mM KCl, 15 mM NaCl, 5 mM $MgCl_2$, 1 mM $CaCl_2$, 250 mM sucrose, 0.3% NP-40, freshly supplemented with 1 mM DTT and Halt protease and phosphatase inhibitor (Thermo Fisher Scientific)) and incubated on ice for 5 min. Nuclei were collected by centrifugation (600 rcf, 5 min at 4 °C), washed once with nuclear-extraction buffer without NP-40, pelleted again, then resuspended in prechilled hypotonic buffer (3 mM EDTA, 0.2 mM EGTA and freshly supplemented with 1 mM DTT and Halt protease and phosphatase inhibitor) and incubated on ice for 30 min to release chromatin. Chromatin was pelleted for 5 min at 1,700 rcf at 4 °C in a cooled centrifuge and subsequently washed twice with hypotonic buffer. Chromatin pellets were solubilized using probe sonication in lysis buffer 100 mM triethylammonium bicarbonate (TEAB), 1% sodium deoxycholate (SDC), 10% isopropanol, 50 mM NaCl, 1:1,000 Pierce Universal Nuclease (Thermo Fisher Scientific) supplemented with Halt protease and phosphatase inhibitor. The protein concentration was measured using the Quick Start Bradford protein assay (BioRad) according to the manufacturer's protocol. A total of 5 mg of protein with an equal contribution from each individual sample was reduced with 5 mM Tris-2-carboxyethyl phosphine (TCEP) for 1 h, followed by alkylation with 10 mM iodoacetamide for 30 min, then digested by adding trypsin (Pierce) at final concentration 75 ng µl$^{-1}$ to each sample followed by incubation for 18 h at room temperature. For chromatin proteomics, 15 µg of protein digest was taken from each sample and labelled with TMTpro multiplexing reagents (Thermo Fisher Scientific), according to the manufacturer's protocol. SDC was precipitated with formic acid at a final concentration of 2% (v/v) and centrifuged for 5 min at 10,0000 rpm. Supernatant containing TMTpro-labelled peptides were dried with a centrifugal vacuum concentrator. The remaining peptides were cleaned up using Pierce Peptide Desalting Spin Columns (Thermo Fisher Scientific), and then dried using a speed vacuum. Acetylated peptides were enriched with the PTMScan HS Acetyl-Lysine Motif (Ac-K) Kit (Cell Signalling Technologies, 46784) according to the manufacturer's instructions, dried using a speed vacuum, resuspended in 100 mM TEAB and labelled with TMTpro according to the manufacturer's protocol. Acetyl-enriched peptides were fractionated using the Pierce High pH Reversed-Phase Peptide Fractionation Kit (Thermo Fisher Scientific, 84868) according to the manufacturer's protocol, dried using a speed vacuum and resuspended in 0.1% trifluoroacetic acid (TFA). Before MS analysis of the chromatin proteome, TMTpro-labelled peptides were fractionated with high-pH reversed-phase (RP) chromatography using the Waters XBridge C18 column (2.1 mm × 150 mm, 3.5 µm) on the Dionex UltiMate 3000 high-performance liquid chromatography (HPLC) system. Mobile phase A was 0.1% ammonium hydroxide (v/v) and mobile phase B was 100% acetonitrile and 0.1% ammonium hydroxide (v/v). Peptide separation was performed with a gradient elution of 200 µl min$^{-1}$ with the following steps: isocratic for 5 min at 5% phase B, gradient for 40 min to 35% phase B, gradient to 80% phase B in 5 min, isocratic for 5 min, and re-equilibrated to 5% phase B. The fractions were collected in a 96-well plate every 42 s to a total of 65 fractions, then concatenated into 12 fractions, dried and reconstituted in 0.1% TFA. The samples were analysed using a Real Time Search-SPS-MS3 method on the Orbitrap Ascend mass spectrometer coupled to a Dionex UltiMate 3000 system. From each fraction, an estimated amount of 3 µg of peptides per fraction was injected onto a C18 trapping column (Acclaim PepMap 100, 100 µm × 2 cm, 5 µm, 100 Å) at a flow rate of 10 µl min$^{-1}$. The samples were processed via a 120 min low-pH gradient elution on a nanocapillary reversed-phase column (Acclaim PepMap C18, 75 µm × 50 cm, 2 µm, 100 Å) at 50 °C. MS1 scans were collected from the 400–1,600 $m/z$ range in the Orbitrap with the following settings: resolution, 120,000; AGC, standard; injection time, auto; and including 2–6 precursor

charge states. Dynamic exclusion was set to 45 s, repeat count of 1, mass tolerance of 10 ppm and the exclude isotope option was enabled. MS2 spectra were acquired in the ion trap at Turbo scan rate, HCD collision energy was set to 32% and 35 ms maximum-injection time was allowed. MS2 scans were searched against the human canonical and isoforms database (UniProt, 16 December 2022) using the Comet search engine in real time with the following filters: tryptic peptides with maximum of 1 missed cleavages, static modifications included Cys carbamidomethylation (+57.0215) and N-terminal/Lys TMTpro (+304.2071), variable modifications Asn/Gln deamidation (+0.984) and Met oxidation (+15.9949), with maximum of variable modifications set to 2; close-out was enabled with a maximum of 4 peptides per protein. Precursors matching these criteria were selected for SPS10-MS3 scans performed at an orbitrap resolution of 45,000 with the normalized HCD collision energy set to 55%, AGC set at 200% and 200 ms maximum injection time. Acetyl-enriched peptides were analysed using an MS2-HCD method with collision energy set to 35%, AGC set at $1 \times 10^5$ and 105 ms maximum injection time. The SequestHT and Comet search engines were used to analyse the acquired spectra in Proteome Discoverer v.3.0 (Thermo Fisher Scientific) for protein identification and quantification. For analysis of the chromatin proteome, the precursor mass was set to 20 ppm and fragment mass tolerance was 0.5 Da. Spectra were searched for fully tryptic peptides with a maximum of two missed cleavages. N-terminal/Lys TMTpro and carbamidomethyl at Cys were defined as static modifications. Dynamic modifications included oxidation of Met and deamidation of Asn/Gln. For peptides enriched for acetylated lysine, the precursor mass was set to 10 ppm and the fragment mass tolerance was set to 0.02 Da. Spectra were searched for fully tryptic peptides with a maximum of three missed cleavages. N-terminal TMTpro and carbamidomethyl at Cys were defined as static modifications, while dynamic modifications included oxidation of Met, deamidation of Asn/Gln, and TMTpro or acetyl at Lys. Peptide confidence was estimated using the Percolator node. Peptide FDR was set at 1% and validation was based on $q$ value and a target–decoy database search. Spectra were searched against reviewed UniProt human protein entries. The reporter ion quantifier node included a TMTpro quantification method with an integration window tolerance of 15 ppm and an integration method based on the most confident centroid peak at the MS3 or MS2 level. Only unique peptides were used for quantification, considering protein groups for peptide uniqueness. Peptides with an average reporter signal-to-noise ratio of >3 were used for quantification. For the chromatin proteome, the data were normalized to total loading at the proteome level, whereas, for the respective acetylome, the data were corrected for loading for acetylated peptides only. Relative abundances were calculated by dividing normalized protein/peptide abundances by the average abundance of all TMTpro channels per biological replicate.

## Immunoblotting and immunofluorescence

Cells were scraped from dishes in 2× SDS buffer (120 mM Tris-HCl pH 6.8, 4% SDS, 20% glycerol). After total protein quantification, equal protein amounts were run on 4–12% Bis-Tris NuPAGE precast gels, transferred to nitrocellulose membrane (GE Healthcare) and immunoblotted with the indicated antibodies. For chromatin fractionation, cells were washed with cold PBS and resuspended in CSK buffer (10 mM PIPES pH 7.0, 100 mM NaCl, 300 mM sucrose, 3 mM MgCl₂, protein inhibitor cocktail (Roche, EDTA-free, 1 tablet per 10 ml), EGTA-free phosphatase inhibitors (1 mM NaF, 0.7 mM β-glycerol phosphate, 0.2 mM Na₃VO₄, 8.4 mM Na₄P₂O₇), 0.7% Triton X-100), incubated on ice for 30 min and centrifuged at 20,000$g$ for 10 min at 4 °C. The supernatant (soluble fraction) was collected and maintained on ice. The pellet was washed twice with cold PBS and sonicated (four pulses of 10 s at 30% amplitude with 10 s resting on ice between cycles) in CSK buffer. The protein concentration of soluble and chromatin fractions was determined using the Bradford assay and Laemmli buffer was added to the samples. Finally,

the samples were boiled, centrifuged at 16,000$g$ for 1 min and equal amounts were loaded onto SDS–PAGE gels. For immunofluorescence studies, cells on coverslips were fixed in a formaldehyde lysis solution (4% formaldehyde, 0.5% Triton X-100, 1× PBS), washed with 1× PBS and permeabilized in 0.5% Triton X-100, 1× PBS. Blocking was performed in 1× PBS, 0.1% Triton X-100, 10% FBS for 1 h, followed by incubation with primary or secondary antibodies in the same solution. Washes were performed using 1× PBST (1× PBS, 0.1% Triton X-100). Coverslips were mounted in Vectashield Mounting Medium with DAPI (Vector Laboratories, H1200-10). Images were collected on the Leica SP8 with ×63/1.4 NA oil objectives, using the Leica Application Suite X software (LAS-X). Images were deconvolved using Huygens Professional v.19.04 software (Scientific Volume Imaging); processing and analysis were performed using ImageJ v.1.53a and Adobe Illustrator 2021. All of the images shown are the projections of $z$ optical sections.

## SIRT1 inhibition assays

Cells were preincubated with EX 527 (selisistat; SIRT1i; Selleckchem) resuspended in DMSO or with DMSO alone for 3 days and then seeded at a density of $2.5 \times 10^5$ cells per 10 cm dish, maintaining either SIRT1i or DMSO in the culture medium. The next day, cells were treated with tamoxifen or mock treated for 24 h. The number of living cells at each timepoint was determined after trypsinization using the Countess II machine (Life Technologies). To determine the dose of tamoxifen that resulted in full depletion of *DSCC1* in the hTERT RPE-1 *DSCC1*$^{Δ/flox}$*cre*$^{tam}$ cell line, cells were grown in the presence of different concentrations of the compound. After 3 days of tamoxifen treatment (Sigma-Aldrich), cell survival was observed by staining with crystal violet (Sigma-Aldrich; 1% aqueous solution). The dose that killed all *DSCC1*$^{Δ/flox}$*cre*$^{tam}$ cells, but not WT hTERT RPE-1 control cells (100 nM), was used for subsequent experiments (Extended Data Fig. 7). To determine the dose of SIRT1i that fully inhibits SIRT1 activity in cultured hTERT RPE-1 cells, the acetylation of p53 at Lys382, a bona fide SIRT1 substrate[86], was examined. Cells were grown in the presence of different concentrations of SIRT1i for 3 days (Extended Data Fig. 9). To avoid interference from other histone deacetylases 5 µM vorinostat (Sigma-Aldrich) was added to the cells 2 h before gamma irradiation (5 Gy). Then, 3 h later, the samples were collected and acetylation of p53 at Lys382 was examined using western blotting.

## SIRT1 in vitro deacetylation assay

These experiments were performed in *HDAC8*-KO HAP1 cells (Horizon Discovery) and also hTERT RPE-1 cells (Extended Data Fig. 9). p53 (a known SIRT1 target) was purified 5 h after gamma irradiation (10 Gy) of hTERT RPE-1 cells that were previously treated with 10 µM selisistat and 5 µM vorinostat (SAHA; Sigma-Merk, SML0061). SMC3 was purified from exponentially growing *HDAC8*-KO HAP1 cells. Both proteins (p53 and SMC3) were purified by immunoprecipitation as follows: cell pellets were resuspended in lysis buffer (50 mM Tris-Cl pH 8, 150 mM NaCl, 1 mM EDTA, 0.5% igepal, complete EDTA-free protein inhibitor cocktail from Roche and phosphatase inhibitor cocktails 2 and 3 from Sigma-Aldrich) and quantified. For p53, 2 mg of protein from hTERT-RPE-1 cell lysates was incubated with 20 µl of Dynabeads (protein G) and 6 µl of anti-p53 antibodies. For SMC3, 10 mg of HAP1 cell lysate was incubated with 40 µl of Dynabeads (protein A) and 12 µg of anti-SMC3 antibodies. Both incubations were performed overnight at 4 °C. The next morning, the beads were washed four times with cold lysis buffer and twice with reaction buffer (50 mM Tris-HCl, pH 7.5, 150 mM NaCl, 1 mM MgCl₂) and resuspended in 50 µl of reaction buffer. For the deacetylation reaction, 10 µl of beads was incubated with 1 µl of human recombinant SIRT1 (Sigma-Aldrich) in a total volume of 30 µl of reaction buffer supplemented with 1.5 µM NAD⁺. The reactions were incubated for 3 h at 30 °C with shaking. Finally, the reactions were stopped by the addition of 10 µl of 4× Laemmli sample buffer and incubation at 95 °C for 5 min. The samples were then immunoblotted with the respective antibodies.

## siRNA experiments in RPE-1 cells

A total of 200,000 RPE-1 $DSCC1^{Δ/flox}cre^{tam}$ cells were seeded per well of a six-well plate and allowed to attach overnight. The cells were then transfected with either non-targeting (referred to as SCR control) or targeting siRNA at 25 nM using 5 μl DharmaFECT 1 transfection reagent (Horizon Discovery T-2001-02) according to the manufacturer's instructions. After 24 h, the medium was replaced in all wells and cells were treated with or without 100 nM 4-OHT. Cells were incubated for a further 5 days before collecting and cell counting by trypan blue exclusion. A list of all siRNAs used is provided in Supplementary Table 6.

**MN counting in HAP1 cells.** HAP1 cells were seeded at an equal density, grown on coverslips and transfected with siRNAs targeting luciferase or *DSCC1*. All siRNAs were ON-TARGETplus SMARTpools manufactured by Dharmacon and used at a final concentration of 20 μM per siRNA. Transfections were performed using Invitrogen RNAiMAX (Life Technologies) according to the manufacturer's instructions. Transfections were repeated after 48 h. After an additional 24 h, the coverslips were fixed with freshly prepared 3.7% paraformaldehyde in PBS for 7 min at room temperature. Cells were permeabilized and stained for 10 min with 0.1% Triton X-100 in PBS, supplemented with 1 μg ml$^{-1}$ DAPI at room temperature. The coverslips were washed once with PBS, and mounted onto glass slides using Prolong Gold (Invitrogen). The slides were imaged and deconvolved on the THUNDER Imager (Leica Microsystems) and analysed using ImageJ (v.2.1.0/1.53k). A cell was scored as harbouring MN when the nucleus had one or more MN in its proximity. At least 400 cells were scored per condition.

**Analysis of cohesion defects in RPE-1 TP53-KO and RPE-1 TP53/DSCC1-double-KO cells.** RPE-1 *TP53/DSCC1*-double-KO cells were generated as previously reported[51] and were cultured in DMEM + 8% FCS. For analysis of cohesion defects, cells were incubated for 20 min with 200 ng ml$^{-1}$ demecolcine (Sigma-Aldrich), collected, incubated for 20 min in 0.075 M KCl and fixed in 3:1 methanol:acetic acid. Cells were washed in fixative three times, dropped onto microscopy slides and stained with 5% Giemsa (Merck). For each condition, cohesion defects were counted in 50 metaphases on two coded slides.

## Statistics and reproducibility

Statistical analyses were performed using Prism (v.9.1.0/v.10.1, GraphPad) or R (v.3/v.4.3.1). All statistical details are provided in the figure legends. All experiments were performed independently at least three times, and were replicated by independent researchers using multiple models and using blinding where possible. *T*-tests were unpaired.

## Reporting summary

Further information on research design is available in the Nature Portfolio Reporting Summary linked to this article.

## Data availability

The CRISPR screen data have been deposited to the European Nucleotide Archive under accession number ERP105493. The MS proteomics data have been deposited at the ProteomeXchange Consortium via the PRIDE[87] partner repository with the dataset identifiers PXD034902, PXD030499 and PXD045110. All other data are available in the Supplementary Information or Source Data. All mouse phenotyping data are available at the IMPC website (www.mousephenotype.org) and at GitHub. Source data are provided with this paper.

## Code availability

All code used is available at GitHub (https://github.com/team113sanger/Large-scale-analysis-of-genes-that-regulate-micronucleus-formation/).

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

**Acknowledgements** We thank all of the members of the D.A. and G.B. laboratories for discussions; J. Forment for help with the quantification of genomic instability in primary MEFs; J. Coates for helping with cellular assays in the S.P.J. laboratory; A. Apetrei and S. Coates for helping with the MN screen while doing undergraduate research; P. Jallepalli for providing the *DSCC1* conditional RPE-1 cells; and the members of the International Mouse Phenotyping Consortium. The Sanger Mouse Genetics Project was supported by Wellcome (098051); P.A.C. by a Cancer Research UK program grant (C3/A18795); and N.A.K. by the National Institutes of Health grant U54 HG006370-01. The data analysed by J.R.B.P. and K.A.K. were obtained from the UK Biobank Resource under application 9905. J.R.B.P. and K.A.K. were supported by the Medical Research Council (unit programs: MC_UU_12015/2, MC_UU_00006/2). Research in the Jackson laboratory was funded by a Cancer Research UK Discovery award (DRCPGM\100005), Wellcome Investigator Award (06388/Z/17/Z) and ERC Synergy Award DDREAMM (855741). Gurdon Institute core infrastructure funding was provided by Cancer Research UK (C6946/A24843) and Wellcome (WT203144). S.P.J. receives his salary from the University of Cambridge. M.J.A. and D.J.A. are supported by Cancer Research UK (C20510/A21717), Wellcome (WT206194), The Medical Research Council and the ERC (319661). The G.B. laboratory is supported by the UK Dementia Research Institute, which receives contributions from UK DRI, the UK MRC, the Alzheimer's Society and Alzheimer's Research UK, as well as a grant from the Romanian Ministry of Research, Innovation and Digitization (no. PNRR-III-C9-2022-I8-66; contract 760114).

**Author contributions** G.B. and D.J.A. conceived the study and wrote the manuscript with assistance from all of the authors. R.E.M. developed the MN assay screening pipeline and screened mutant mouse lines together with G.B., A.B., C.D.R.-E., R.V., N.G. and M.d.C.V.-H. R.E.M. identified the *Dscc1* mutant, generated the *Dscc1* mutant cohorts for characterization/ tumour watch (including collection of tissues for pathology) and generated *Dscc1* mutant embryos for imaging and derivation of fibroblasts with help from L.v.d.W. M.J.A. performed histopathological analyses. A.B., N.A.T., G.T. and M.R. performed and analysed the CRISPR screen. A.B. validated the results of the CRISPR screen in human iPS cells. A.H. validated the results of the screen in HEK293 human cells. I.S. validated the results of the screen and performed molecular analyses in the human RPE-1-derived cell lines. The Sanger Mouse Genetics Project, A.O.S., L.v.d.W. and C.J.L. performed the mouse pipeline phenotyping. N.A.K. performed the statistical analyses of the MN screen and was involved in the statistical analyses for the mouse phenotypic pipeline. Y.Z. and R.O.-L. analysed the phenotypic data. W.J.W., T.J.M. and S.H.G. performed the phenotyping of the *Dscc1* embryos. P.A.C. performed the analysis of cohesion defects in iPS cell cultures. B.F. and F.Y. performed the chromosomal analyses. J.H. cultured the KD of *Dscc1* in iPS cells and helped with mouse analyses. B.B., H.Z., L.Y. and J.C. performed the MS analyses. J.R.B.P. and K.A.K. performed the LOY GWAS analyses. B.B., V.H., J.R.V., N.K., A.G.H., M.S.v.R., I.S., B.D.R., L.B., M.A.R., J.d.L. and H.Z. generated and analysed functional studies. M.S.v.R. and B.D.R. performed the analyses of *WAPL/PDS5A* in HAP1 cells.

L.B., J.R.V., H.Z. and A.G.H. performed the arrayed MN screen in human cells. S.P.J. obtained grant funding for various aspects of the work, was involved in planning aspects of the study, and interpreting data. D.J.A. and C.D.R.-E. performed the COSMIC and DD GWAS analyses.

**Competing interests** A patent on the repurposing of SIRT1 inhibitors has been filed by the University of Cambridge. The data presented in this patent are included in the main paper and Supplementary Information. D.J.A. is a consultant for Microbiotica and Ono Therapeutics and receives research funding from Astra Zeneca and OpenTargets. M.R. is employed by Artios Pharma and is also a shareholder. J.R.B.P. is an employee of Insmed Innovation UK and holds stock/stock options in Insmed, and also receives research funding from GSK. The other authors declare no competing interests.

**Additional information**
**Correspondence and requests for materials** should be addressed to D. J. Adams or G. Balmus.

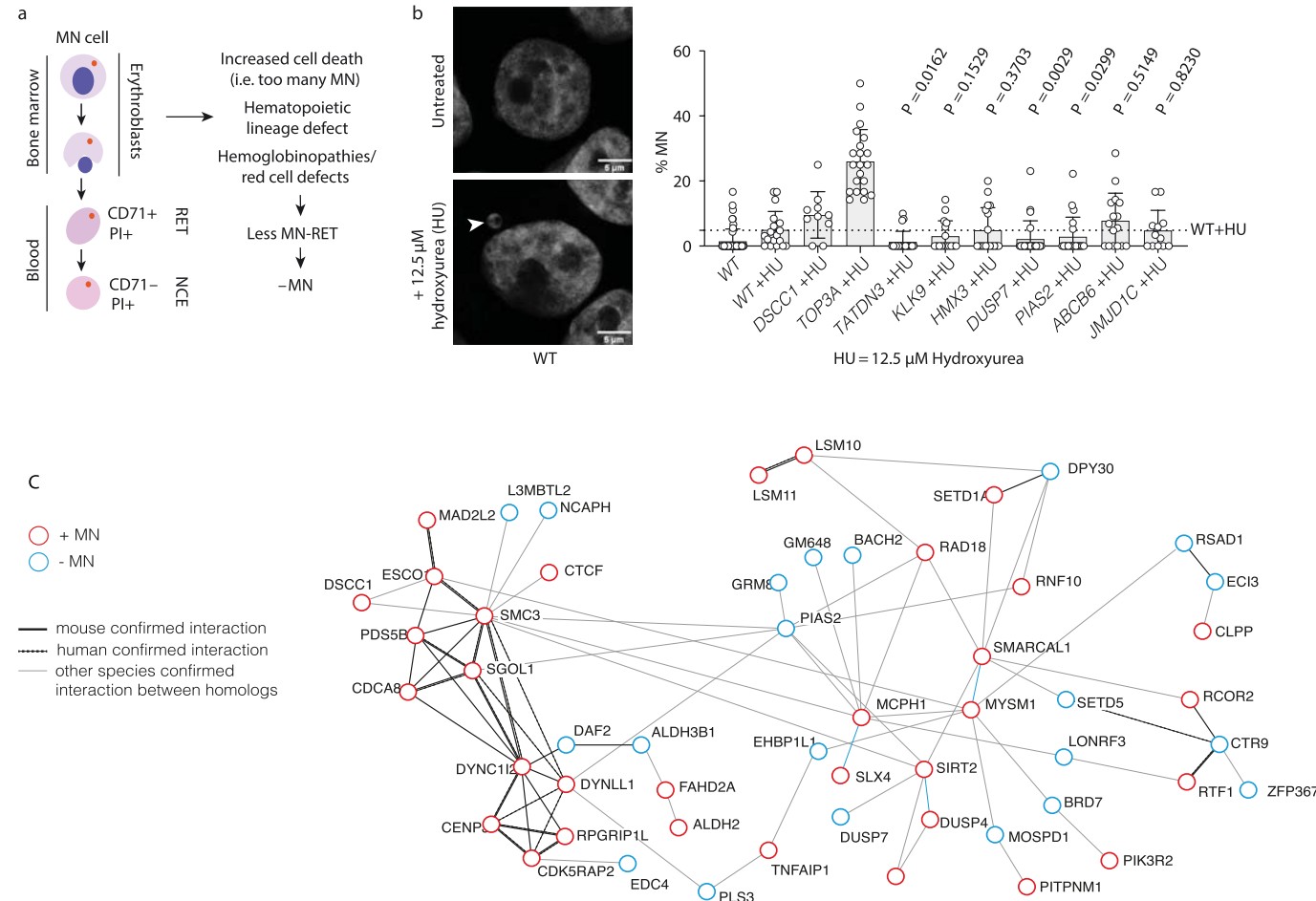

**Extended Data Fig. 1 | Micronucleus formation in mice links multiple genetic determinants. a**, Wild-type (WT) male mice have, on average, 0.2% erythrocytes containing micronuclei; lines with a significant drop in this % were scored as showing decreased micronuclei (-MN) (see Source Data for raw values). While the +MN category likely denote lines with increased genomic instability, alternative explanations could explain the -MN lines including haemoglobinopathies, haemopoietic lineage defects or a profound increase in genomic instability in the erythroblasts that would not allow such cells to reach the peripheral circulation because of cell death prior to release from the bone marrow. **b**, Validation of seven -MN Tier 1 genes in human CHIP-212 cells; KO of *DSCC1* and *TOP3A* were used as positive controls. Genes were disrupted using CRISPR-Cas9 gRNAs (Supplementary Table 6). MN levels were induced using 12.5 μM hydroxyurea for 3 days. Left panels show representative DAPI-positive and control nuclei. The arrow points to a MN. Significance was assessed using a Mann-Whitney U (two-sided) test. For each gene, data were collected from 3 independent wells (which were treated as biological replicates) by randomly selecting >200 cells and manually counting micronuclei. Bars represent mean with error bars s.d. **c**, Interactome analysis using STRING v.11[88] and BioGRID v.4.4[89] showed that 54/145 of the protein products of the genes we identified as affecting MN formation have been reported to interact, thus building a core 'MN network'.

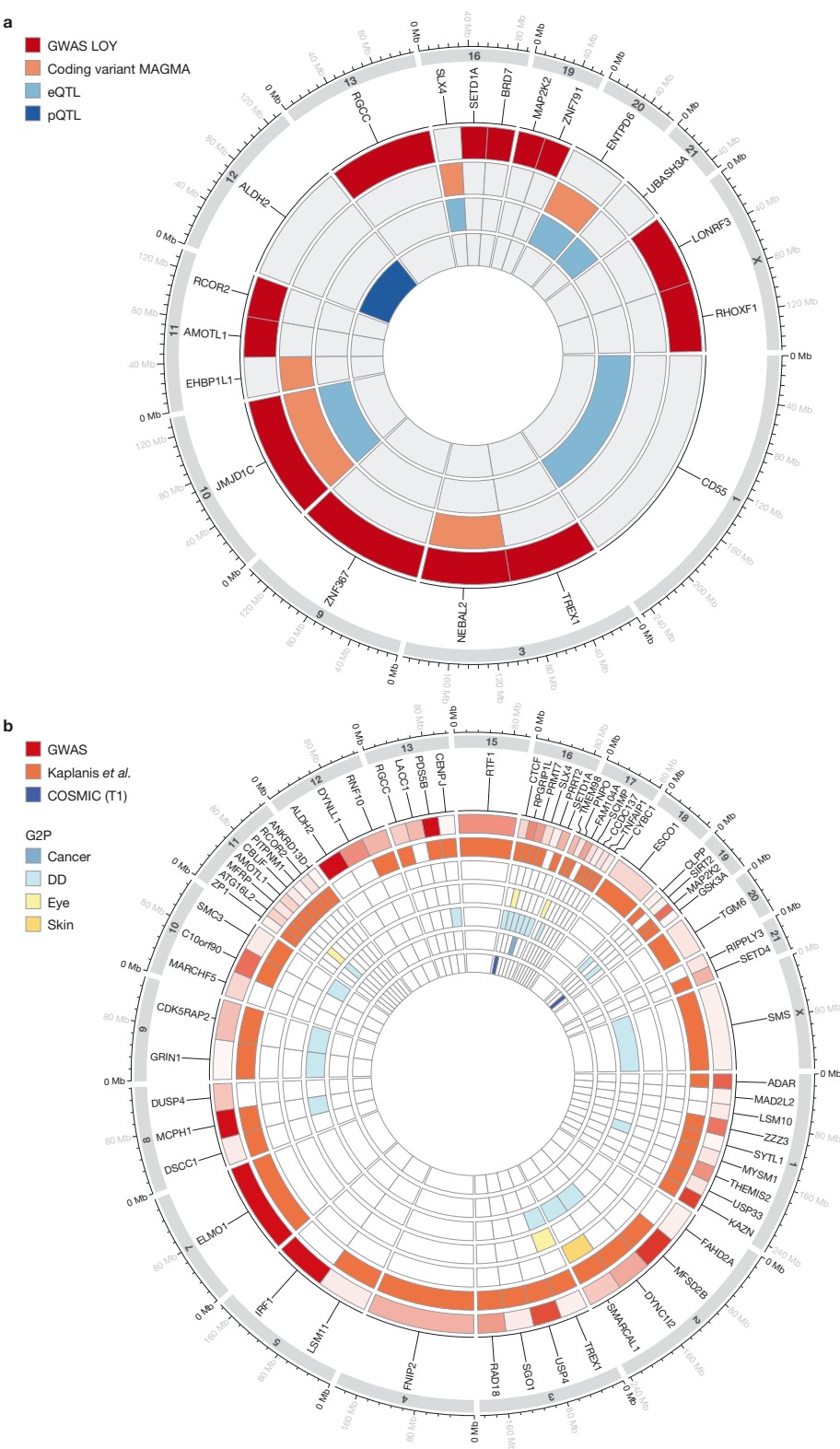

**Extended Data Fig. 2** | See next page for caption.

**Extended Data Fig. 2 | Integration of MN hits with human genome-wide association (GWAS) and other genetic studies. a**, Circos plot showing the overlap between the mouse MN genes and human genetic datasets. The outer concentric circle (red) indicates whether the listed gene is proximal to a signal in the loss-of-Y (LOY) GWAS[26]. The orange circle indicates whether there was a significant gene-level association between LOY and gene variants with predicted deleterious effects. Light blue indicates co-localization between GWAS and mRNA levels in the blood for the listed gene and dark blue, equivalently for blood protein levels. Corresponding results can be found in Supplementary Table 2. **b**, Circos plot showing the overlap between the +MN genes identified in mice and human disease datasets. Corresponding results can be found in Supplementary Table 3. G2P: genes to phenotypes[29]. GWAS: GWAS Catalog. COSMIC (T1): COSMIC Tier 1 cancer genes[30]. Kaplanis et al., ref. 31; DD, developmental disorders (see Methods). In the outer ring the "redness" denotes the number of associations with genes in the GWAS catalogue. Where there are multiple genes on a chromosome, we segmented the chromosome into equal bins.

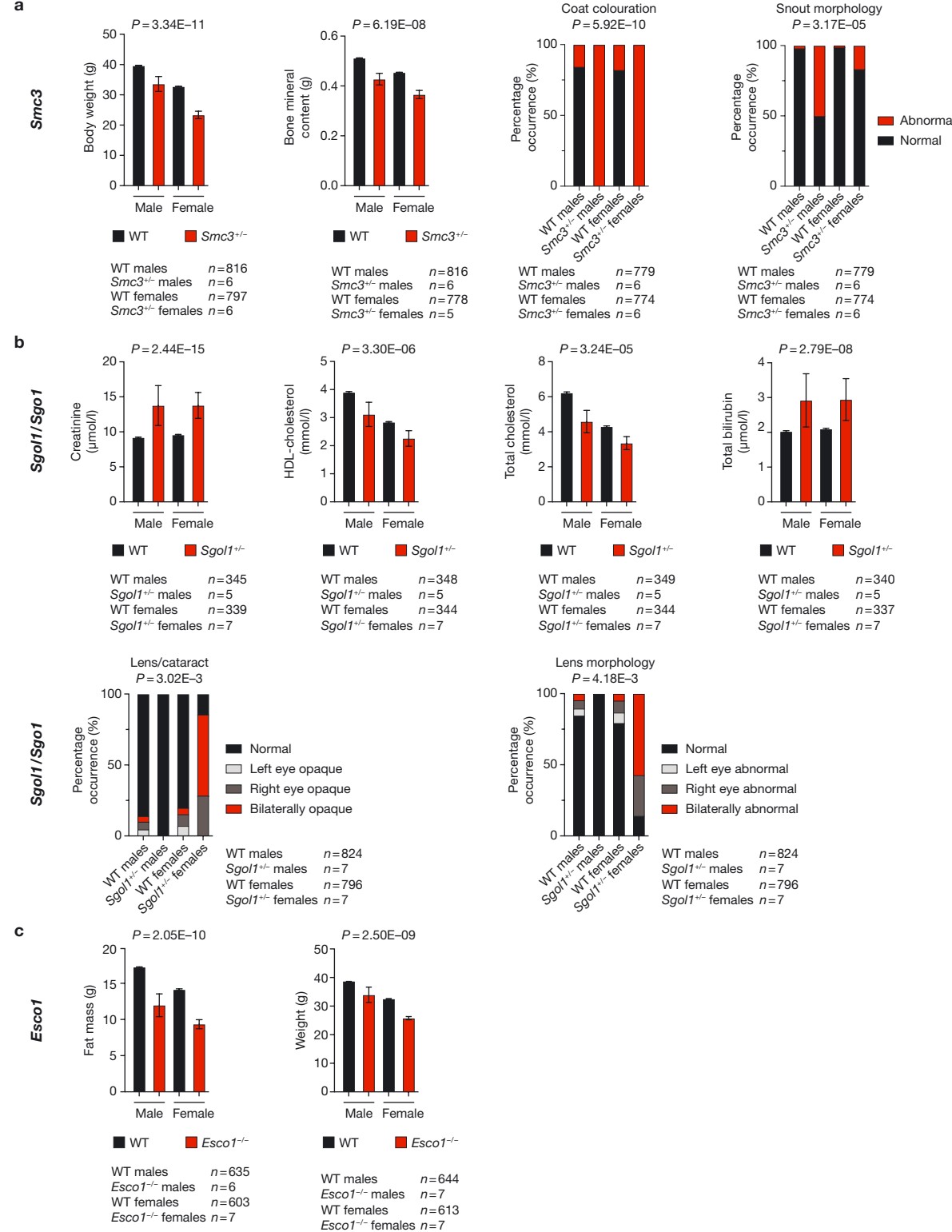

**Extended Data Fig. 3 | Phenotypic analysis of mouse mutants related to cohesion defects.** Bar graphs showing selected significant phenotypic differences for the cohesin-related mouse lines *Smc3* (**a**), *Sgol1/Sgo1* (**b**) and *Esco1* (**c**). The individual mouse was considered the experimental unit within the studies. The mean is shown with error bars denoting the s.d. The numbers of mice and statistical methods used are detailed on the IMPC website[59]. P values were calculated depending on the data type (continuous or categorical) within the Phenstat Package in R (version 3.18) which deploys linear mixed modelling (10.18129/B9.bioc.PhenStat) or using Fisher's tests (for categorical data). These statistical methods are available in the phenotyping file on Github/Figshare and are also available on the IPMC database website[59]. The data presented are a snapshot from September 2023 (see Methods) as part of IMPC release 19.

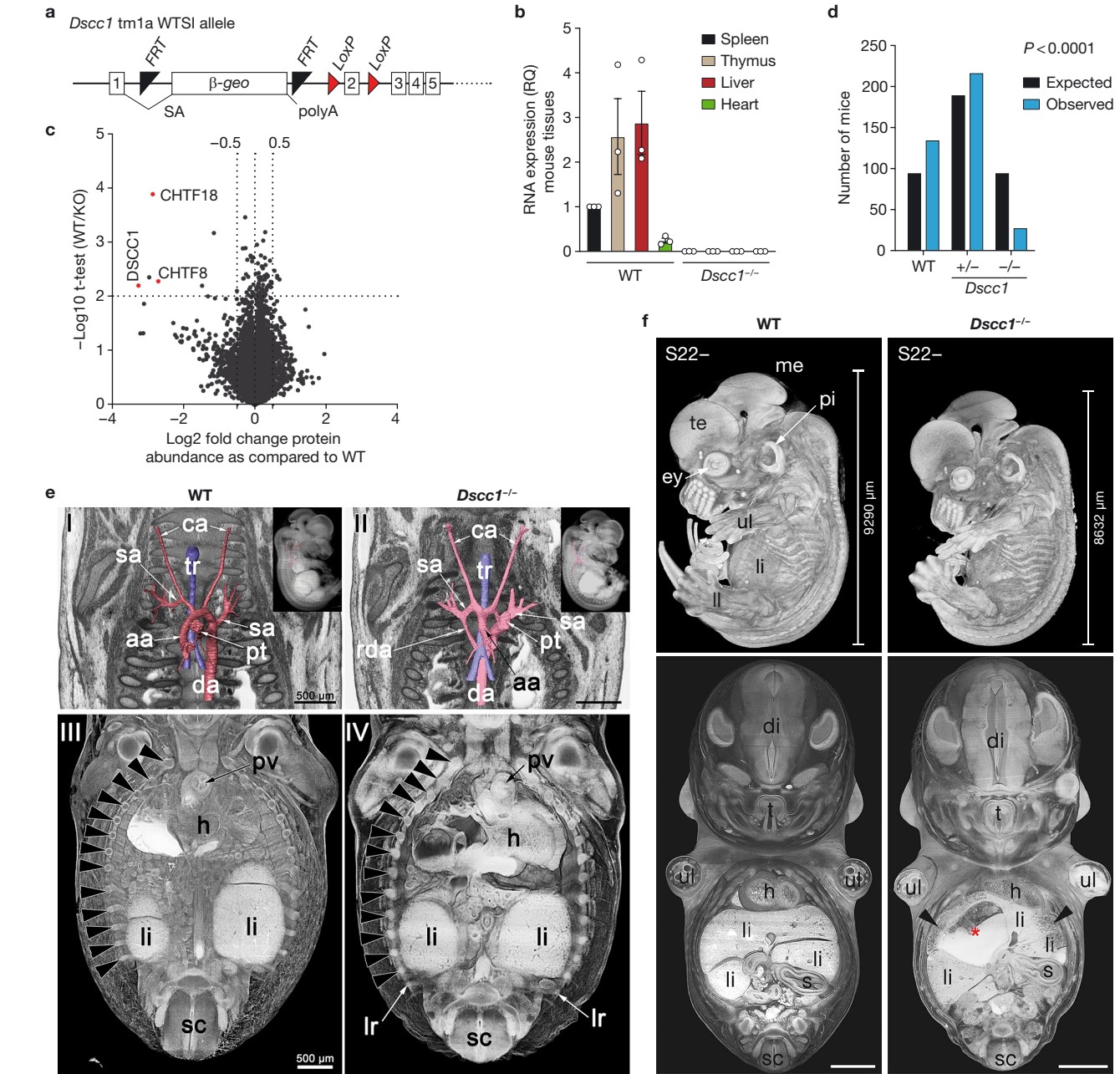

**Extended Data Fig. 4** | See next page for caption.

**Extended Data Fig. 4 | *Dscc1* mutant mice show cardiac and vascular anomalies and reduced viability. a**, Diagram shows the targeting of the *Dscc1* locus on mouse chromosome 15. A beta-galactosidase gene-trap including a splice acceptor site (SA) and a polyadenylation sequence (polyA) were inserted in intron 1 of the *Dscc1* gene. Further elements were inserted to allow the generation of a conditional allele, such as *FRT* and *LoxP* sites. **b**, Bar graph showing quantitative PCR analysis of *Dscc1* transcripts in adult mouse tissues. n = 3 mice with n = 5 technical replicates each. Mean is plotted with error bars representing s.e.m. **c**, Mass-spectrometry analysis of E13.5 embryo heads showing depletion of DSCC1, CHTF8 and CHTF18 proteins (members of the DSCC1-CHTF18-CHTF8 protein complex). The raw files were processed with Proteome Discoverer 2.4 (ThermoFisher) using the Sequest HT search engine and the analysis is presented in Source Data. Proteins/peptides were validated using Percolator. Only unique peptides were used for quantification. Red dots denote key significantly differentially expressed proteins (Student's two-tailed t-test was used to determine significance). Two embryos of each genotype were analysed in this way. **d**, Mice born from *Dscc1* heterozygous (+/−) intercrosses that survived past post-natal day 10 (P10) were genotyped and a Chi-squared analysis (two-tailed) was performed using the expected versus observed numbers of each genotype. Approximately a third of the expected *Dscc1*[−/−] mice survived past P10. **e**, Skeletal and vascular abnormalities in *Dscc1*[−/−] (right panels) embryos and for comparison control (left panels) embryos. Great intrathoracic arteries at developmental stage S22- (upper panels) are shown. Abnormal persistence of right dorsal aorta (rda) in a *Dscc1*[−/−] embryo. Surface models of the arteries in front of a coronal section through HREM data from anterior. Inlay shows the surface models inside a semitransparent volume model from right. Coronally sectioned semi-transparent volume models of thorax and abdomen from ventral (lower panels). The regular 13 ribs are indicated with arrowheads. Note the lumbar rib (lr) in the *Dscc1*[−/−] embryo. **f**, Growth delay and liver abnormalities in *Dscc1*[−/−] embryos. Control/wild-type (left panels) and *Dscc1*[−/−] (right panels) embryos are shown. *Upper row:* Growth and developmental delay can be seen in a E14.5 *Dscc1*[−/−] embryo relative to WT embryo. In addition, the developmental stage (S22-) of *Dscc1*[−/−] mutants is earlier than of wild-type littermates and as expected from reference data[75]. *Lower row:* Abnormal liver. Coronally sectioned semi-transparent volume models of thorax and abdomen from ventral. Blood filled cyst (red asterisk) and enlarged liver sinusoids (arrowheads). te, telencephalon; me, mesencephalon; ey, eye; pi, pinna; ul, upper limb; ll, lower limb; li, liver; tr, trachea; ca, common carotid artery; h, heart; pv, pulmonary valve; sa, subclavian artery; aa, ascending aorta; da, descending aorta; pt, pulmonary trunk; rda, right descending aorta, di, diencephalon; t, tongue; s, spleen; sc, spinal cord. For this experiment, n = 3 embryos/genotype were analysed.

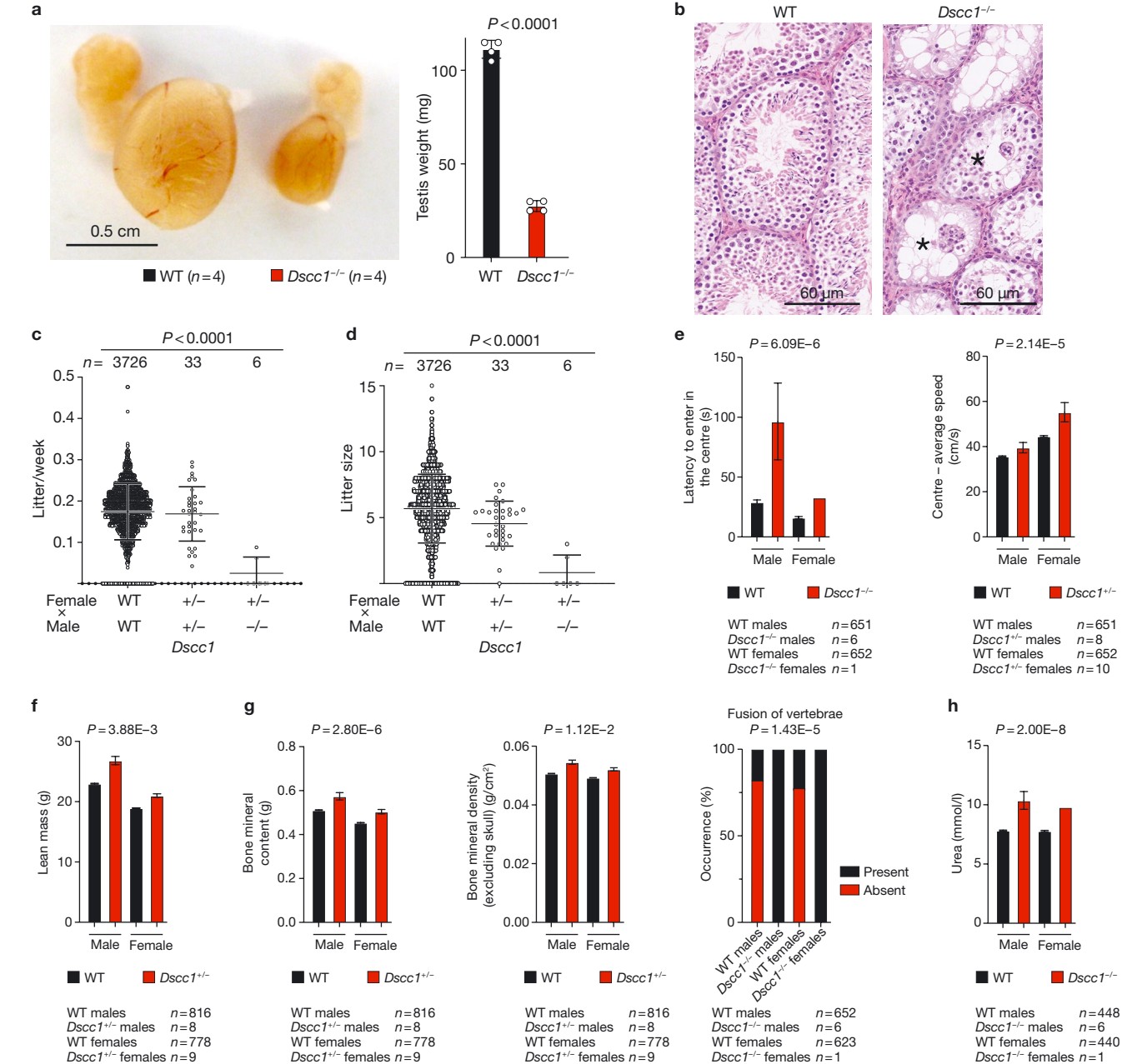

**Extended Data Fig. 5 | Surviving *Dscc1* mutant mice show phenotypes affecting several major organ systems. a**,**b**, *Dscc1⁻/⁻* male mice have smaller testes than wild-type (WT) littermates. **a**, Macroscopic image for comparison is shown (left). Scale bar shows size estimate. Testis weight (right). Significance was assessed using a Student's two-tailed t-test. Data are from four mice (37 weeks old) with average weight per mouse testis shown. Bars and whiskers are mean and s.d. **b**, *Dscc1⁻/⁻* testis showed complete agenesis of seminiferous tubules (asterisk) following haematoxylin and eosin staining. Scale bar shows size estimate. In this experiment, n = 3 animals per genotype were analysed. **c**,**d**, Breeding using *Dscc1* mutant mice and WT controls and quantification of the number and size of litters produced showed that *Dscc1⁻/⁻* male mice are sub-fertile. Significance was assessed using a student's two-tailed t-test.

Bars represent mean with s.d. The n numbers are shown in the figure. Limited matings were performed with *Dscc1⁻/⁻* female mice but these animals produced live born pups. Owing to the reduced penetrance of *Dscc1⁻/⁻* mice, elements of the phenotyping were performed using *Dscc1⁺/⁻* mice (as indicated). **e**–**h**, *Dscc1* mutant mice show significant differences in lean mass, skeletal structure and development, behaviour and metabolism. Bars represent mean with s.d. P values were calculated depending on the data type (continuous or categorical) within the Phenstat Package in R (version 3.18) which deploys linear mixed modelling (10.18129/B9.bioc.PhenStat) or using Fisher's tests (for categorical data). These statistical methods are available in the phenotyping file on Github/Figshare and are also available on the IPMC database website[59]. The data presented are a snapshot from September 2023 (see Methods) as part of IMPC release 19.

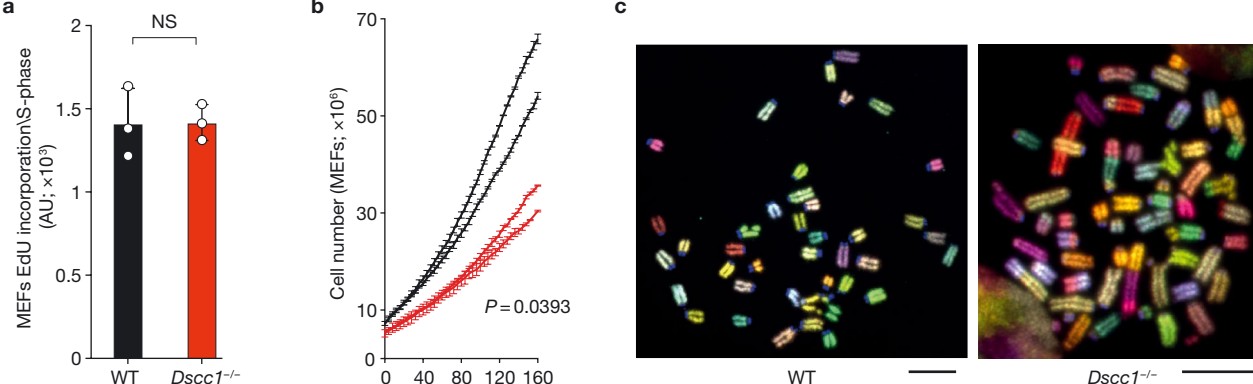

**Extended Data Fig. 6 | *Dscc1* mouse embryonic fibroblasts (MEFs) grow slower and show increased genomic instability. a**, Bar graph quantifying the incorporation of 5-Ethynyl-2'-deoxyuridine (EdU) in MEFs of the indicated genotypes shows no difference in the S-phase cell cycle fraction. Significance was assessed using a Student's two-tailed t-test. Experiment performed three independent times (n = 3 biological replicates). Mean is shown with the error bars denoting s.d. **b**, Cell growth profiles (left) for cell lines derived from the same E13.5 litter show that *Dscc1^-/-^* MEFs grow significantly slower than wild-type (WT) controls. For **a**,**b**, two independent WT and two independent *Dscc1^-/-^* MEF lines were derived from littermate embryos; n = 3 independent wells/replicates each. Bars represent mean with s.e.m. Significance was assessed using an Student's two-tailed t-test comparing the area under the curve (AUC). **c**, Fluorescent in-situ hybridization (FISH) images of metaphases from MEFs derived from littermates showing increased chromosomal aberrations characteristic in *Dscc1^-/-^* cells. This experiment was replicated three independent times. Size bar 10 μm. This image is a lower magnification of the image shown in Fig. 2d.

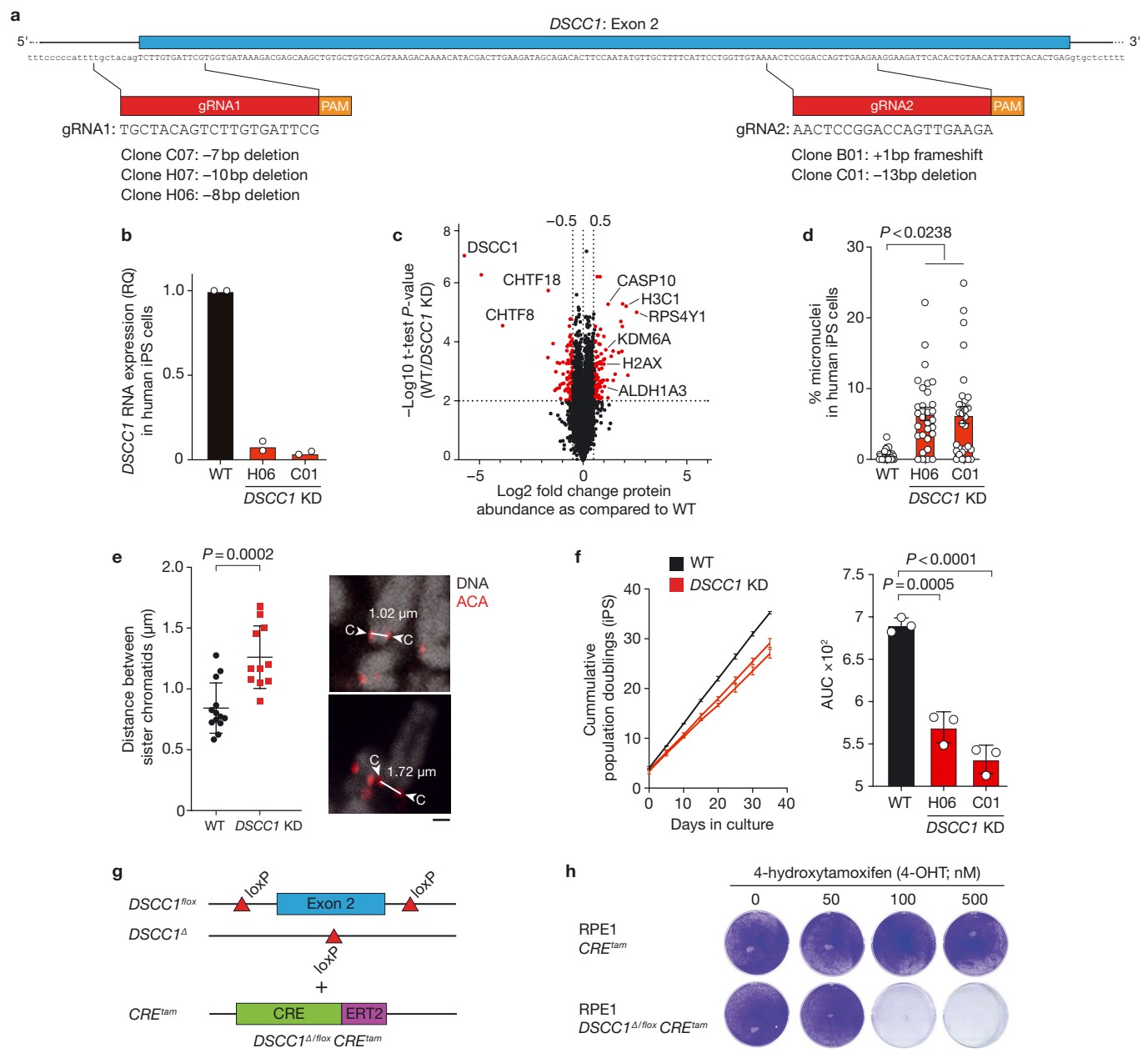

**Extended Data Fig. 7** | See next page for caption.

**Extended Data Fig. 7 | *DSCC1* mutant induced pluripotent stem (iPS) cells show increased micronucleus formation. a**, Diagram showing the targeting of the *DSCC1* locus on human chromosome 8. Two different guide RNAs (gRNA) were used to generate clones. The position of these gRNAs is shown. After transfection, individual clones were picked and genotyped (see Methods). **b**, Bar graph of the *DSCC1* transcript levels in two independent iPS cell lines by quantitative PCR analysis using TaqMan probes shows effective gene knock-down (KD) and transcript truncation. n = 2 independent experiments with n = 5 technical replicates each. **c**, Mass-spectrometry analysis of *DSCC1* KD iPS cells showing depletion of DSCC1 levels as well as disruption of other DSCC1-CHTF18-CHTF8 complex proteins. Red dots denote peptides that are significantly changed in abundance compared to wild-type (WT) (see Methods). Raw data are presented in Source Data. Significance was assessed by one sample T-test, two-tailed, Benjamini-Hochberg FDR = 0.05. The data are from one mass-spec run comparing a reference proteome of the parental "BOB" iPS line to 5 proteomes from independently cultured *DSCC1* KD clones. **d**, Quantification of the micronucleus levels in two independent iPS *DSCC1* KD clones and isogenic (WT) control cells. MN were measured as DAPI positive structures present outside of the nuclear envelope. n = 3 independent experiments/biological replicates with each dot equalling an independent field of view with >50 cells; Bars represent mean, error bars are s.e.m. Analysis was performed using a Mann-Whitney U (two-sided) test. **e**, Quantification of the inter-centromere (c) distance by use of anti-centromere antibodies (ACA)

shows that loss of *DSCC1* leads to a significant increase in the distance between the two sister chromatids. Each data point is an independent measure randomly selected from across three independent cultures (see Source Data). Bars represent mean with error bars denoting the s.d. Significance was assessed using a Student's two-tailed t-test. Scale bar 1 μm. **f**, Cumulative population doubling analysis over 40 days in culture shows that *DSCC1* KD iPS cells grow significantly slower than isogenic WT control cells (area under the curve, AUC). Data were generated with n = 2 independent lines (H06 [upper] and C01 [lower]) with n = 3 biological replicates each. Bars are means with error bars denoting s.d. Significance was assessed using Student's two-tailed t-test on AUC values. **g**, TERT RPE-1 *DSCC1*$^{\Delta/flox}$ conditional cells were imported as a gift from the Jallepalli Laboratory[46]. In these cells, one allele of *DSCC1* has been disrupted (Δ; delta), while the other allele is flanked by *loxP* sites (flox). To create an inducible system, we stably integrated a tamoxifen inducible CRE recombinase (CRE) construct (where the CRE recombinase is fused to a mutant oestrogen ligand-binding domain (ERT2) that requires the presence of 4-hydroxytamoxifen (4-OHT) for activity; *CRE*$^{tam}$). **h**, Optimal 4-hydroxytamoxifen dose determination by crystal violet staining of hTERT RPE-1 *DSCC1*$^{\Delta/flox}$*CRE*$^{tam}$ cells treated for three days with different 4-hydroxytamoxifen (4-OHT) concentrations. The dose that killed all *DSCC1*$^{\Delta/flox}$*CRE*$^{tam}$ cells, but did not affect their parental hTERT RPE-1 *CRE*$^{tam}$ counterpart, was used in subsequent experiments (100 nM). This experiment was performed *n* = 3 times (biological replicates).

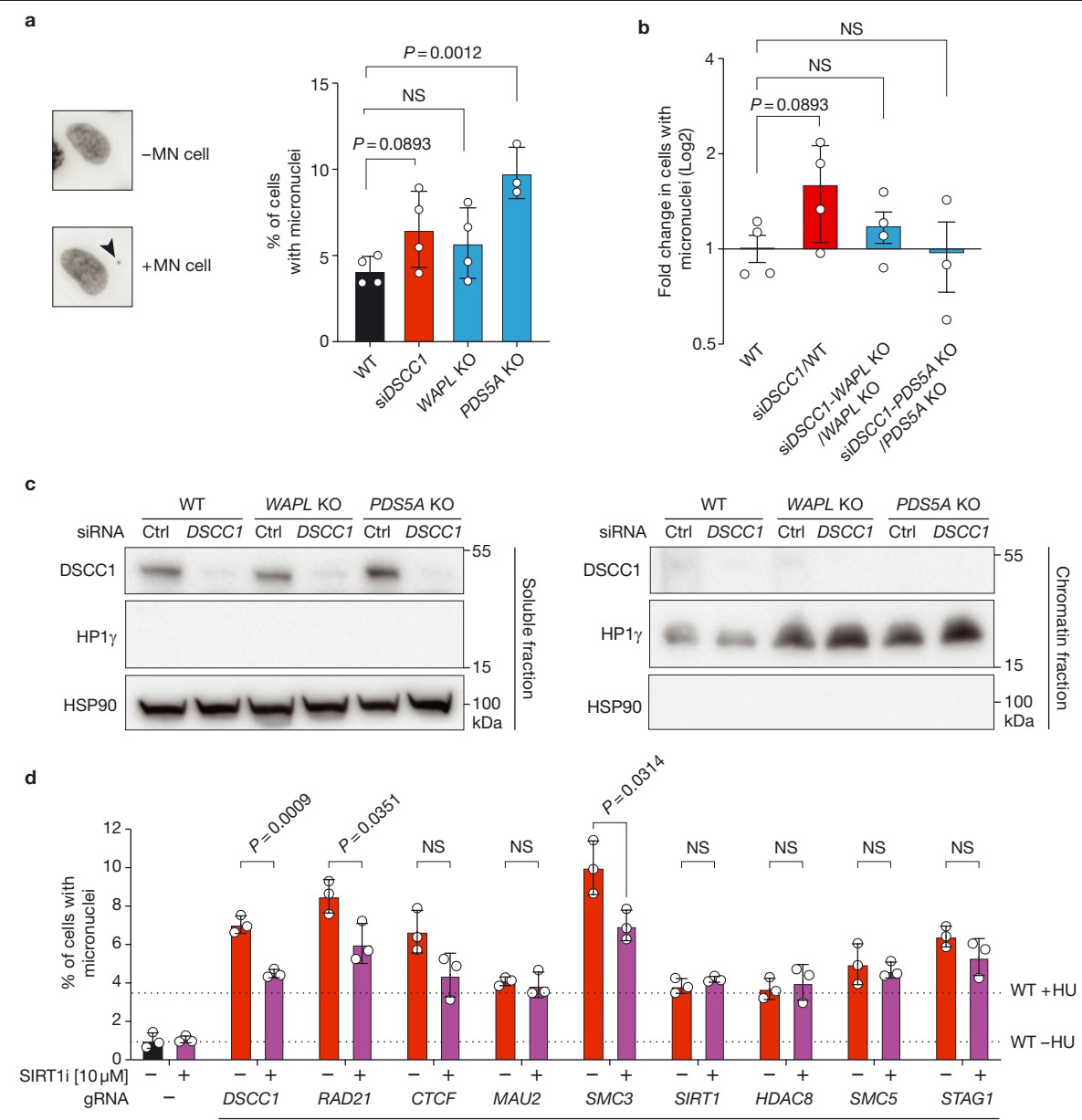

**Extended Data Fig. 8 | Validation of the *DSCC1* suppressor CRISPR screen.**
**a**, Quantification of the % of cells with micronuclei (MN) in HAP1 cells. Depletion of *DSCC1* (*siDSCC1*) or *PDS5A* (KO), but not *WAPL* (KO), resulted in a significant increase in %MN. Each point on the graph represents and independent experiment where more than 50 cells were counted. Representative images are presented on the left hand side; arrow head points at DAPI positive MN. Significance was assessed using a two-tailed Student's t-test. n = 3 biological replicates with n ≥ 50 cells counted each. Bars represent mean with s.d.
**b**, Quantification of the fold-change in MN formation in *siDSCC1*/WT as compared to *siDSCC1-WAPL* KO and *siDSCC1-PDS5A* KO relative to *WAPL* KO and *PDS5A* KO alone, respectively (HAP1 background). Significance was assessed using a two-tailed Student's t-test (NS, not significant; *P* > 0.05). n = 3 biological replicates with n ≥ 50 cells counted per replicate. Bars represent mean with s.d.

**c**, Representative western blot images from soluble and chromatin fraction extracts from HAP1 cells depicting siRNA depletion of DSCC1. This experiment was repeated three times and the uncropped images are presented in Supplementary Fig. 2. **d**, Quantification of the effect of SIRT1 inhibition with Selisistat (EX 527; SIRT1i; 10 μM) on the MN formation when the cohesion-associated genes *RAD21, CTCF, MAU2, SMC3, HDAC8, SMC5* and *STAG1* were disrupted using CRISPR-Cas9 (see Methods) in RPE-1 cells. *DSCC1 KO* and *SIRT1 KO* were used as controls. To increase the dynamic range, MN were induced by a 3-day chronic treatment with 50 μM hydroxyurea (HU) (see HU titration for the different cell lines in Supplementary Table 7). Significance was assessed using Student's two-tailed t-test (NS, not significant; *P* > 0.05). n = 3 biological replicates with n ≥ 50 cells per replicate counted. Bars represent mean with s.d.

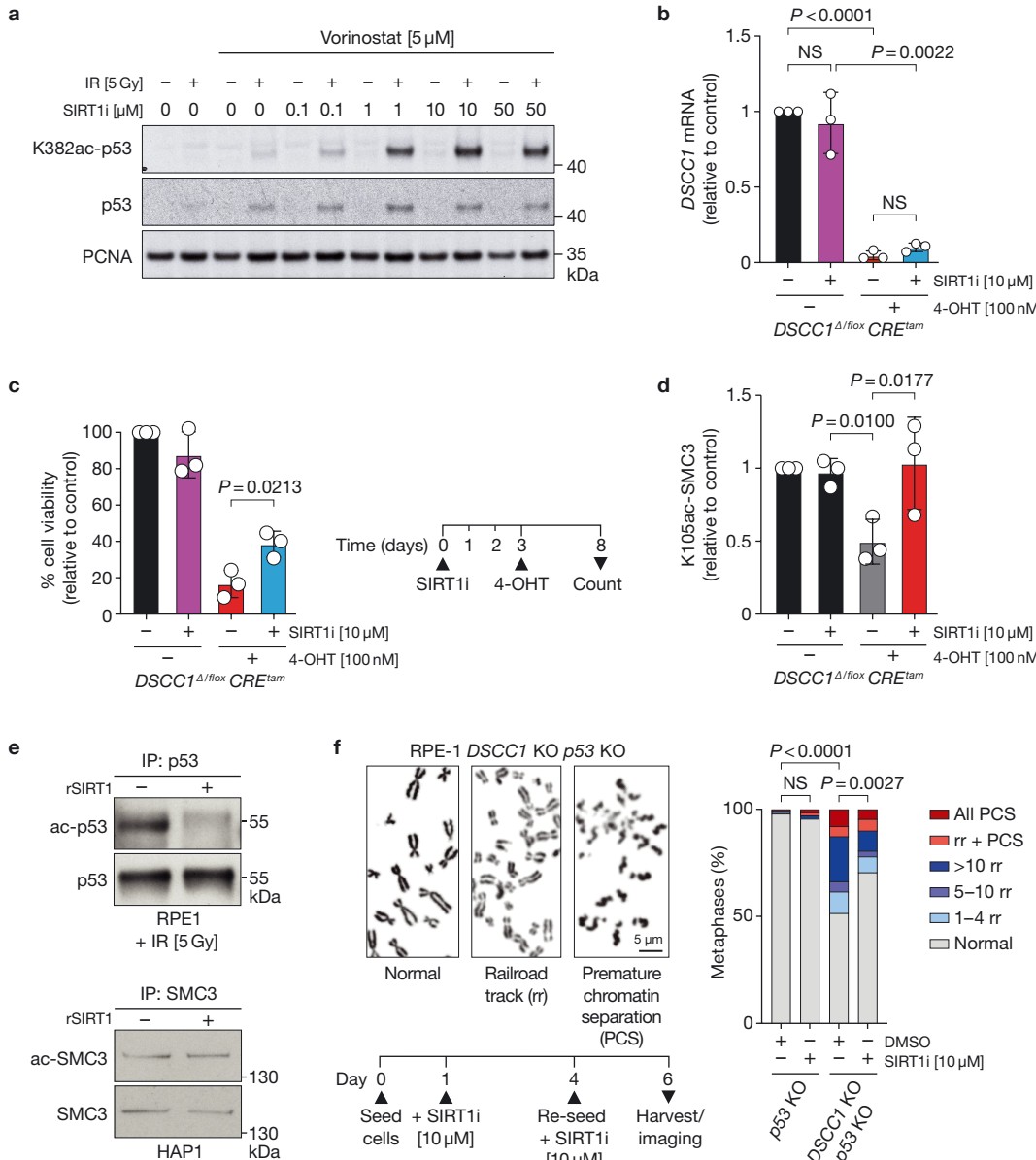

**Extended Data Fig. 9 | SIRT1 inhibition rescues the DSCC1 cohesion defect independent of p53 and not via direct SMC3 deacetylation. a**, Representative images of immunoblots showing the effect of increasing concentrations of SIRT1 inhibitor on the p53-K382 acetylation levels in the RPE-1 $DSCC1^{\Delta/\text{flox}}$ $CRE^{tam}$ cell line upon gamma irradiation in the presence of the HDAC1 inhibitor, vorinostat. Uncropped western images are presented in Supplementary Fig. 2. This experiment was performed once. **b**, $DSCC1$ mRNA quantification by RT-qPCR in the RPE-1 $DSCC1^{\Delta/\text{flox}}$ $CRE^{tam}$ cell line after the indicated treatments show that $DSCC1$ can be depleted by the addition of 100 nM 4-OHT. Note that SIRT1 inhibition (SIRT1i) does not significantly affect $DSCC1$ levels. n = 3 independent experiments/biological replicates with n = 5 technical replicates each. Bars represent mean with s.d. Significance was assessed by a Student's two-tailed t-test (NS, not significant; $P > 0.05$). **c**, Independent experiment in the RPE-1 $DSCC1^{\Delta/\text{flox}}$ $CRE^{tam}$ cell line in the presence of SIRT1 inhibitor upon DSCC1 depletion by 4-OHT treatment shows that SIRT1i can significantly rescue cell viability. Significance calculated using a two-tailed Student's t-test. The experiment was repeated three independent times (biological replicates) with three technical replicates each. Mean is plotted with the error bars denoting the s.d. **d**, Quantification of the western blots for which representative images are presented in Fig. 4d showing SMC3 acetylation at K105 is significantly

restored in the $DSCC1^{\Delta/\text{flox}}$ $CRE^{tam}$ cells upon SIRT1i. Statistical analysis was performed using a two-tailed Student's t-test; bars represent mean with s.d. The experiment was performed three times independently. **e**, Representative immunoprecipitation (IP) followed by immunoblotting from a SIRT1 in vitro deacetylation assay performed by using recombinant SIRT1 protein (rSIRT1). The rSIRT1 can deacetylate p53 at K382 (upper panels) but cannot deacetylate SMC3 even in absence of HDAC8 (lower panels). n = 3 independent repeats (biological replicates). **f**, On the left, representative images of metaphase chromosomes from three independent experiments/biological replicates illustrating normal, railroad (RR) chromosomes as well as chromosomes with premature sister chromatid separation (PCS) in different stages from TERT-RPE-1-$p53$ KO cells as compared to TERT-RPE-1 $p53$ KO $DSCC1$ KO with and without SIRT1i. Size bar 5 μm. Below is represented the timeline for the experimental setup. On the right, quantification of the different RR and PCS events in the metaphasis from RPE-1 $p53$ KO vs. RPE-1 $p53$ KO $DSCC1$ KO with and without SIRT1i. The experiment was repeated n = 3 independent times (biological replicates). More than 50 metaphases/genotype were analysed. Statistical analysis comparing the proportions of normal cell metaphases and cell-defect metaphases was performed with a logistic regression model[90]; NS, not significant; $P > 0.05$.

# Reporting Summary

## Statistics

For all statistical analyses, confirm that the following items are present in the figure legend, table legend, main text, or Methods section.

| n/a | Confirmed | |
|---|---|---|
| ☐ | ☒ | The exact sample size (*n*) for each experimental group/condition, given as a discrete number and unit of measurement |
| ☐ | ☒ | A statement on whether measurements were taken from distinct samples or whether the same sample was measured repeatedly |
| ☐ | ☒ | The statistical test(s) used AND whether they are one- or two-sided<br>*Only common tests should be described solely by name; describe more complex techniques in the Methods section.* |
| ☐ | ☒ | A description of all covariates tested |
| ☐ | ☒ | A description of any assumptions or corrections, such as tests of normality and adjustment for multiple comparisons |
| ☐ | ☒ | A full description of the statistical parameters including central tendency (e.g. means) or other basic estimates (e.g. regression coefficient) AND variation (e.g. standard deviation) or associated estimates of uncertainty (e.g. confidence intervals) |
| ☐ | ☒ | For null hypothesis testing, the test statistic (e.g. *F*, *t*, *r*) with confidence intervals, effect sizes, degrees of freedom and *P* value noted<br>*Give P values as exact values whenever suitable.* |
| ☒ | ☐ | For Bayesian analysis, information on the choice of priors and Markov chain Monte Carlo settings |
| ☐ | ☒ | For hierarchical and complex designs, identification of the appropriate level for tests and full reporting of outcomes |
| ☐ | ☒ | Estimates of effect sizes (e.g. Cohen's *d*, Pearson's *r*), indicating how they were calculated |

*Our web collection on statistics for biologists contains articles on many of the points above.*

## Software and code

Policy information about availability of computer code

| Data collection | No commercial software of this kind was used. |
|---|---|
| Data analysis | As stated in the manuscript we used Prism software for statistical analysis (Version 9.1.0 & 10.1). Analysis of the screen results used software written in R (glmmTMB, version: 1.0.1). https://github.com/glmmTMB/glmmTMB. R V4 was used. FlowJo V7-10 was used. |

For manuscripts utilizing custom algorithms or software that are central to the research but not yet described in published literature, software must be made available to editors and reviewers. We strongly encourage code deposition in a community repository (e.g. GitHub). See the Nature Portfolio guidelines for submitting code & software for further information.

## Data

Policy information about availability of data

All manuscripts must include a data availability statement. This statement should provide the following information, where applicable:

- Accession codes, unique identifiers, or web links for publicly available datasets
- A description of any restrictions on data availability
- For clinical datasets or third party data, please ensure that the statement adheres to our policy

Data Availability Statement
The CRISPR screen data have been deposited to the European Nucleotide Archive with accession number ERP105493. The mass spectrometry proteomics data have been deposited to the ProteomeXchange Consortium via the PRIDE partner repository with the dataset identifiers, PXD034902, PXD030499 and PXD045110. All other data is available in the Supplementary or Source Data of the paper.

International Mouse Phenotyping Consortium Database release V19 was used. This is available via the IMPC website (www.mousephenotype.org) and also via the github page for the project.

# Field-specific reporting

Please select the one below that is the best fit for your research. If you are not sure, read the appropriate sections before making your selection.

☒ Life sciences ☐ Behavioural & social sciences ☐ Ecological, evolutionary & environmental sciences

For a reference copy of the document with all sections, see nature.com/documents/nr-reporting-summary-flat.pdf

# Life sciences study design

All studies must disclose on these points even when the disclosure is negative.

| | |
|---|---|
| Sample size | Statistics and reproducibility<br>Prism (version 9.1.0 & 10.1, GraphPad) were used to perform the statistical analysis, unless otherwise described. All statistical details are provided in the Figure legends. Significance is expressed as P values (NS, not significant; $P>0.5$; $*P<0.05$, $**P<0.01$, $***P<0.001$ and $****P< 0.0001$).<br>n>3; all data-points presented in the figures.<br><br>For most experiments we had no a priori way of determining sample size so we used empirical evidence/knowledge to estimate the effect size/biologically meaningful outcome. We did this by following the guide by Karen Grace-Martin (https://www.theanalysisfactor.com/sample-size-estimation-without-past-reliable-pilot-data-or-evidence/) and Simon Bates: https://www.nc3rs.org.uk/3rs-resources/how-decide-your-sample-size-when-power-calculation-not-straightforward. |
| Data exclusions | no exclusion |
| Replication | all experiments repeated n=3 times (or more). i.e. biological replicates. Data from all replicate experiments was included except where the experiment failed for technical reasons (such as a failed PCR, western blot that didn't transfer etc.). |
| Randomization | For animal experiments randomization was performed by Mendelian inheritance. For other experiments we did not randomize but used pre-defined conditions for data acquisition so as to avoid any bias. |
| Blinding | Blinding of mouse experiments was not always possible as the genotype was on the cage card but samples for FACS analysis were barcoded so the genotype of the mouse was not available to the experimenter unless they explicitly looked it up. Similarly for CRISPR screening we made no prior assumptions about what hits would be return from our whole genome screens. Blinding was not possible for other experiments such as transfections. |

# Reporting for specific materials, systems and methods

We require information from authors about some types of materials, experimental systems and methods used in many studies. Here, indicate whether each material, system or method listed is relevant to your study. If you are not sure if a list item applies to your research, read the appropriate section before selecting a response.

## Materials & experimental systems

| n/a | Involved in the study |
|---|---|
| ☐ | ☒ Antibodies |
| ☐ | ☒ Eukaryotic cell lines |
| ☒ | ☐ Palaeontology and archaeology |
| ☐ | ☒ Animals and other organisms |
| ☒ | ☐ Human research participants |
| ☒ | ☐ Clinical data |
| ☒ | ☐ Dual use research of concern |

## Methods

| n/a | Involved in the study |
|---|---|
| ☒ | ☐ ChIP-seq |
| ☐ | ☒ Flow cytometry |
| ☒ | ☐ MRI-based neuroimaging |

## Antibodies

| | |
|---|---|
| Antibodies used | CD71-FITC antibody (SouthernBiotech, cat. no. 1720-02; 0.5 mg/ml; 1:500)18; SIRT1 antibody (Rabbit Cell Signalling #2496S; 1:1000); anti-centromere antibody (Antibodies Inc, #15-234-0001, 1:1000); anti-rabbit Alexa 488 (ThermoFisher, #A11034, 1:2000); goat anti-human Alexa 647 (ThermoFisher, #A21445, 1:2000). DSCC1 (H0079075-B01P, Novus Biologicals; 1:1000); HSP90 (F-8, Santa Cruz; 1:10.000); HP1-gamma (05-690, Millipore; 1:1000); goat-anti-mouse-PO (DAKO; # P044701; 1:2000). SMC3 antibody (Abcam, AB 9263, 1:250); SMC3 antibody (Thermo Fisher, A300-060A;1:1000,); Anti-acetyl SMC3 mouse antibody (Sigma-Aldrich, #MABE1073, clone 21A7, Lys105/106, LOT: 385016; 1:1000); p53 (Cell Signaling Technology, clone 1C12, #2524S); anti-acetyl p53 (p53-K382Ac; Abcam, ab75754; clone EPR358(2) to p53 acetyl K382; 1:1000); Anti-phospho-Histone H2A.X (Ser139) antibody (clone JBW301; |

Sigma-Aldrich,#05-636-I; 1:1000); beta-actin antibody (Merck, #A5441; 1:10000; 5% milk); GAPDH antibody (clone 6C5; Abcam, ab8245; 1:1000); p21 antibody (Abcam, #ab109520; 1:1000).

| Validation | The monoclonal CD71-FITC (SouthernBiotech, cat. no. 1720-02) antibody was validated by the manufacturer (and by us) by staining of red cells by flow cytometry and characteristic staining of mouse 18-81 cells. |

SIRT1 antibody - we validated this antibody by showing loss of protein expression in a SIRT1 KO line by western blotting.

DSCC1 antibody - validated by the western blotting of DSCC1 KO cell lines (which had been validated by mass-spec).

The anti-centromere antibody (Antibodies Inc, #15-234-0001) was validated by staining of TERT BJ cells (by the manufacturer) and was raised from the serum of a CREST patient. We used the antibody to stain human cells seeing the characteristic staining of centromere.

HSP90 antibody - published >700 times. Validated by knockdown of HSP90 by the manufacturer. https://www.scbt.com/p/hsp-90alpha-beta-antibody-f-8

SMC3 antibody - https://www.thermofisher.com/antibody/product/SMC3-Antibody-Polyclonal/A300-060A - staining, WB

Anti-acetyl SMC3 mouse antibody - https://www.merckmillipore.com/GB/en/product/Anti-acetyl-SMC3-Antibody-Lys105-106-clone-21A7,MM_NF-MABE1073?ReferrerURL=https%3A%2F%2Fwww.google.com%2F . Validated by manufacturer, by IP, WB, staining pattern.

p53 antibody - https://www.cellsignal.com/products/primary-antibodies/p53-1c12-mouse-mab/2524 - WB, staining of cells physiological response. Cited by manufacturer.

anti-acetyl p53 - https://www.abcam.com/products/primary-antibodies/p53-acetyl-k382-antibody-epr3582-ab75754.html . WB: HepG2 cell lysates treated with etopside and TSA. ICC/IF: HeLa cells Flow Cyt (intra): HepG2 cells.

Anti-phospho-Histone H2A.X (Ser139) antibody - https://www.sigmaaldrich.com/GB/en/product/mm/05636?gclid=CjwKCAiAvJarBhA1EiwAGgZl0ETywgkpooS5lhMjBRiEs990yF4OaHxJm8SkY7esNObF8Dfdb4kKoRoCM64QAvD_BwE - physiological response to DNA damage by us and the manufacturer.

beta-actin antibody - https://www.sigmaaldrich.com/GB/en/product/sigma/a5441 - validated by staining of cells, WB, tissue staining.

GAPDH antibody - https://www.abcam.com/products/primary-antibodies/gapdh-antibody-6c5-loading-control-ab8245.html - validated by localisation, size characteristic staining.

p21 antibody - https://www.abcam.com/products/primary-antibodies/p21-antibody-epr362-ab109520.html - validated by KO of p21.

## Eukaryotic cell lines

Policy information about cell lines

| Cell line source(s) | Mouse embryonic fibroblasts (MEFs) were prepared from E13.5 embryos, following timed matings between Dscc1+/− mice. Briefly, embryos were dissected from the decidium, mechanically disrupted and cultured in DMEM supplemented with 10% fetal bovine serum, 1.0 mML-glutamine, 0.1 mM minimal essential medium nonessential amino acids, 100 μg/ml streptomycin sulfate and 100 U/ml penicillin. The initial plating was defined as passage zero (p0), and cells were subsequently maintained on a standard protocol. SIRT1 KO HEK293 cells were obtained from Kerafast (ENH131-FP). Cells were grown in DMEM, 10% FBS, 1% Penicillin-Streptomycin, 1%GlutaMAX. All other cell lines were from ATCC and STR profiled/validated and certified mycoplasma free. HEK293FT cells were from Thermo Fisher. |

| Authentication | HEK293 - SIRT1 KO cells - by WB for loss of SIRT1. Previously also confirmed by PCR of the target locus. |
|  | HEK293FT - STR profiling (manufacturer). Antibiotic resistance and the ability to package lentiviruses. |
|  | iPS cells - validated by whole exome sequencing. |
|  | Mouse Fibroblasts - by genotyping of the targeted alleles in embryos. |
|  | RPE1 DSCC1 and p53 lines - STR profiling (manufacturer/ATCC) and also PCR of the targeted/mutant alleles. |
|  | CHP-212 - STR profiling. |

| Mycoplasma contamination | All cell lines were mycoplasma tested and found to be free of this contaminant. |

| Commonly misidentified lines (See ICLAC register) | No commonly misidentified cell lines were used in the study |

## Animals and other organisms

Policy information about studies involving animals; ARRIVE guidelines recommended for reporting animal research

| Laboratory animals | Mouse; We maintained most of the mice on a pure inbred C57BL/6N background (representing 73% of the mice tested in this study), or for early lines on mixed C57BL/6 backgrounds (e.g., C57BL/6N;C57BL/6BrdTyrc-Brd). For the C57BL/6N background, a core colony was set up using mice from Taconic Biosciences, which was refreshed at set generational points (typically 10 generations) and cryopreserved at regular intervals to avoid genetic drift. All of the ages are provided in the figure legends. With the exception of tumour watch mice all other animals were between 8-15 weeks old. |

| Wild animals | No wild animals were used in this study. |
| Field-collected samples | No field collected samples were used in the study. |
| Ethics oversight | Mouse studies at the Wellcome Sanger Institute (WSI) were performed in accordance with UK Home Office regulations and the UK Animals (Scientific Procedures) Act of 2013 under UK Home Office licenses. These licenses were approved by the WTSI Animal Welfare and Ethical Review Board. |

Note that full information on the approval of the study protocol must also be provided in the manuscript.

# Flow Cytometry

## Plots

Confirm that:

☒ The axis labels state the marker and fluorochrome used (e.g. CD4-FITC).

☒ The axis scales are clearly visible. Include numbers along axes only for bottom left plot of group (a 'group' is an analysis of identical markers).

☒ All plots are contour plots with outliers or pseudocolor plots.

☒ A numerical value for number of cells or percentage (with statistics) is provided.

## Methodology

| Sample preparation | Briefly, 50 µl of blood was collected from a tail bleed into 300 µl of heparin solution in a 1.5-ml tube and fixed in ice cold methanol at −80 °C in a freezer. Post fixation the samples were washed in bicarbonate buffer and 20 µl of sample was transferred to a 96-deep-well plate (800 µl per well capacity). To each sample 80 µl of CD71-FITC antibody, 7 µl of RNAse and 73 µl of bicarbonate buffer and propidium iodide (PI; 1.0 mg/ml solution Sigma-Aldrich, cat. no. P4864). |
| Instrument | Samples were analysed on a LSRFortessa or Cytomics FC500, Becton Dickinson. |
| Software | FlowJo |
| Cell population abundance | minimum of 100 thousand events for each sample |
| Gating strategy | we have described in detail the strategy on Nature protocols 10 (1), 205-215. We have also provided a figure in the supplementary Data file. |

☒ Tick this box to confirm that a figure exemplifying the gating strategy is provided in the Supplementary Information.

