## [Peer Review File · Nature]

Manuscript Title: Genetic determinants of micronucleus formation in vivo

Reviewer Comments & Author Rebuttals

Reviewer Reports on the Initial Version:

Referees' comments:

Referee #1 (Remarks to the Author):

In this study, Adams et al. sought to identify genetic factors that contribute to the formation of micronuclei (MN) in vivo by screening over 6,000 mice across 997 loss-of-function mutant lines. Among these mutant mice, they identified 145 genes that, when knocked out, can lead to either increased or decreased levels of micronucleation in red blood cells. In addition, the authors demonstrated that a subset of these genes overlapped with loci in humans that are susceptible to mosaic loss of Y chromosome (LOY), a biomarker of genomic instability. Additionally, the authors showed that mutant mice that exhibited increased or decreased micronucleation displayed phenotypes that are associated with a wide range of disorders in different systems/organs. The authors then focused on one of the top hits that increased micronucleation—the gene *Dscc1*. Consistent with previous studies, the absence of *DSCC1* leads to DNA damage, slow cell growth, and senescence. In addition, the authors showed that *Dscc1* KO mice displayed some developmental defects similar to human cohesinopathies. By performing a CRISPR-Cas9 screen, the authors identified four suppressors of the *DSCC1*-deficient phenotype. They then showed that by inhibition of one of the suppressors, *SIRT1*, they could rescue the slow cell growth and DNA damage defects of *DSCC1*-depleted cells. This rescue effect was correlated with the restoration of acetylation of *SMC3*. Overall, this is a comprehensive screen of almost 1000 mouse mutants and identifies around 150 genes associated with MN formation. However, although this is an impressive effort, I do not feel that as currently presented this is enough of a conceptual advance for the general audience of Nature for a number of reasons. This is not the first time that RBC micronucleation was used to screen for genome instability mutations (e.g., the Chaos mouse mutants). More importantly, the screen has not identified a new functional class of mutations that cause micronucleation- it is already known that many cell division defects cause micronucleation. The phenotype of micronucleation is induced by many types of genome instability and this screen therefore overlaps with other genome stability screens. This impacts the general utility of this study as a comprehensive resource. The authors do report a few strains that reduce micronucleation, which could be novel. But there are only a few such mutations, the dynamic range of the phenotype is small, no functional theme seems to emerge, and these mutations are not functionally characterized in this paper. Finally, the findings on *SIRT1* suppressing the micronuclei-promoting mutation in *DSCC1*, which they link to cohesion acetylation, are interesting and could have therapeutic implications. However, this part of the work is not well-developed (for example, there is no biochemical demonstration that *SIRT1* deacetylates cohesion). Additionally, the current evidence suggests that *SIRT1* inhibition suppresses the loss of *DSCC1* specifically and there is no evidence that it has a general role in improving genome stability (for example, if it suppresses other mutants identified in the screen). This aspect of the manuscript could there be developed as an independent study.

Specific comments:

SIRT1's rescue effect on DSCC1-depleted cells is partial. DSCC1 is a component of the RFC18 complex that is involved in DNA replication fork progression. It seems formally possible that this rescue could be explained by the restoration of this function. Depletion of WAPL and PDS5A have been previously shown to rescue the DNA replication defect of DSCC1 KO cells (Ref 44). The authors should determine if depletion of these factors rescues the defects they observe in DSCC1-deficient cells.

The authors demonstrate a correlation between the restoration of acetylation of SMC3 and SIRT1 inhibition in this study. This could be novel and therapeutically important. However, as mentioned above, this aspect of the manuscript is not developed mechanistically. Moreover, the four suppressors that reduce the DSCC1 KO phenotype seem to have widely varied cellular functions and the other three (TGFB2, KIF25 and CARS) seem unlikely to directly increase cohesion function. A better understanding of why these other mutations suppress DSCC1-deficient cells would enhance the confidence in the specificity of the suppression results.

Minor issues:

Fig. 2c, A definition of "γH2AX positive" should be provided.

Fig. 2d, Please add a scale bar to the figure.

Fig. 3e, There are no error bars on the left graph except for the DSCC1 KD cell condition and one point on for the WT condition. Please make this consistent among all conditions.

Fig. 3f, The Y-axis label is confusing. Does it indicate the percentage of iPS cells containing micronuclei? Also, please state what the individual dots represent on the graph.

Fig 3h, The contrast between the Western blot signal of SMC3 and the background is low. It is difficult to determine if the SMC3 levels are comparable between the SIRT1⁺ and SIRT1⁻ conditions at 2 days by eye. We suggest that the authors include a quantification of the ac-SMC3 signal normalized to the SMC3 signal.

Supplemental Fig. 5 a-b, Please add a scale bar to your images.

Supplemental Fig. 6c, These two images appear to be at different magnifications but are shown with the same scale bars. Please clarify.

Referee #2 (Remarks to the Author):

Adams et al describe an integrative analysis of blood samples from ~1,000 mouse LOF mutants to identify strains associated with accumulation of extranuclear micronuclei (MN), a hallmark of genomic instability that is itself associated with a number of disease states. They then compare screen data to Biobank genetic associations and, in a separate effort, follow up with in vivo and in vitro characterization of one hit, *Dscc1*, a gene previously known to be involved in sister chromatid cohesion.

A philosophical issue arising from studies of this sort is how much focus in the manuscript to devote to the large-scale screening and data collection/analysis aspects vs. how much functional and mechanistic characterization to apply to select hits to develop the novel biology and to demonstrate the value of the dataset. In this case a substantial effort to screen ~6,000 mice representing ~1,000 genotypes for phenotypes related to abnormal micronucleus formation is analyzed, yielding a very

convincing and well-controlled data set that is represented by the volcano plot in Figure 1a. The mouse screens identify, among other things, a collection of genes associated with ‘cohesinopathies’ – disorders related to sister chromatid cohesion (SCC), and the top hit is *Dscc1*, which has surprisingly not been previously associated with cohesinopathies.

The authors compare these results to Biobank data, specifically for genes related to mosaic loss of Y chromosome (LOY), in an effort to demonstrate the broader relevance of the mouse screen to human genetic syndromes. The effort is commendable but the results are suspect. What is the “overrepresentation fold change” of intersected hits relative to expectation? A borderline P-value (0.033) across all 136 genes in tiers 1-3 is pretty weak (Supp Table 1a), and notable that only collectively do the hits cross the admittedly arbitrary 0.05 significance threshold. Given that the rest of the paper addresses the role of *DSCC1* gene, this section stands alone with fairly unconvincing associations, and could safely be relegated to supplementary information without loss of impact. The enrichment for GWAS LOY seems well represented in the Circos plot in Fig 1b but the other data sets are unconvincing (related question: what is the meaning of the different arc widths per gene in the Circos plots?). Figure 1d nicely demonstrates how a single cell-level phenotype can present as multiple seemingly independent organism-level phenotypes/disease states.

To pursue *in vivo* validation and characterization of the *Dscc1* hit, the authors then create a *Dscc1* knockout mouse and demonstrate the developmental and phenotypic defects associated with the genotype, including those consistent with human cohesinopathies as well as (likely GIN-driven) increased tumor formation in older mice.

Finally, a human iPS cell line was engineered to be *DSCC1* deficient, through methods that are not easily found, and screened with a CRISPR knockout library to identify genetic modifiers of *DSCC1*-induced reduced proliferation (in comparison with a WT screen). This is an excellent idea whose phenotypic suppressor hits, including *TGFBR2*, *SIRT1*, *KIF25*, and *CARS*, offer an intriguing hypothesis about possible rescue of *DSCC1*-driven genetic disease. However very little information about this screen is included, and whether its overall quality is sufficient to drive confidence in the results is not a conclusion one can derive from the data presented. The type and degree of *DSCC1* “knockdown” is not described. The quality of the CRISPR KO screens in the two cell lines is nowhere present (many metrics exist for such quality scores). The purpose of comparative fitness screens is inherently questionable when, e.g, flow based screens for other markers of GIN might be more appropriate. Finally, the synergistic hits are all related to transcription and translation and both parental and *DSCC1*kd cells should reasonably be expected to be highly sensitive to knockout of these genes.

Nevertheless, the follow-up characterizing *SIRT1* inhibition as a suppressor of *DSCC1*-dependent phenotype holds promise. What is obviously missing is: does *SIRT1* inhibition have any effect *in vivo*? The authors have the mouse models already; to what extent does *Sirt1i* (assuming the drug works in mice) rescue *Dscc1*^{-/-} mice? It is almost as if the authors have reserved a Figure 4 for exactly this experiment.

Referee #3 (Remarks to the Author):

Micronuclei are a type of genome instability event that occurs in pathological states such as cancer. The authors begin with a survey of several hundred mouse lines to discover a set of gene functions that increase and decrease the number of micronuclei. They follow up with genetic and pharmacologic strategies to test hits related to DNA replication and sister chromatid cohesion. Overall the manuscript is a big step toward discovering genes that regulate the formation of micronuclei. The manuscript is well written in a conservative way which I greatly appreciate. The number of experiments is extensive and I do not suggest adding anything new. The authors describe statistics and technical replicates and use them effectively. It is quite curious that so many different mouse phenotypes co-occur with decreased MN AND increased MN.

One suggestion is that the authors clarify their working model for DSSC1, SMC3 acetylation, and SIRTUIN 1 in the formation of micronuclei. Do they think loss of DSSC1 leads to double strand breaks during DNA replication that fail to be repaired and that leads to chromosome fragments that missegregate? Or do they think the main problem is that whole sister chromatids fail to be accurately segregated, either through premature sister separation or lagging chromosomes, due to failure to biorient on the mitotic spindle? There is precedence in the literature that loss of SMC3 function leads to increased DNA damage following S phase, lagging chromosomes, and micronuclei (Yueh et al., 2021, Development). Haarhuis et al., Current Biology 2013, showed that WAPL depleted cells accumulate DNA damage and MN. The authors could draw on the existing literature regarding DNA replication, cohesin and MN to formulate a working model.

One minor comment-the authors state that MN contain acentric chromosome fragments (page 3). However, the DNA in the MN does sometimes have centromeres, so they should consider rephrasing.

Referee #4 (Remarks to the Author):

A. Spectacular study of micronuclei in a very large panel of mouse lines followed by extensive studies to characterize the impact that the most significant gene has on genomic instability and further studies of genes that can rescue phenotypes associated with loss of this gene.

B. Seems highly original and provides some major insights. That said parts of this paper are not well integrated and seem superfluous. Also, it seems to be written as a brief report and given the breadth of analyses a longer format might be better.

C. The quality of the data seems high and there are a very large number of supplementary data files to back up the findings. That said, I found the GWAS of loss of Y to and some of the subsequent studies to be noncontributory. Loss of Y may be due to the same genes that contribute to development of micronuclei but loss of Y can be due to many other mechanisms(<https://www.nature.com/articles/s41576-019-0202-7>). Therefore, I did not understand the relevance of this analysis to the manuscript. Also, looking at Figure 1, the most significant gene involved in micronuclei is DSSC1 as shown in panel A, but panels B and C do not include that gene at all, which indicates a lack of relevance of the LOY analyses to the rest of the paper. I find this component to be distracting. For analysis of DSSC1 performed on the UKBB, I found the description confusing. The animal model for DSSC1 used loss of function variants. I imagine the authors just used

SNPs for the DSSC1 analysis in the UKBB and those results would not be very relevant to the mouse phenotypes. A better approach could be to use the recently released exome sequencing data from UKBB or to use imputed data to identify variants that lead to a nonfunctional variant of DSSC1 and evaluate the phenotypes among individuals with severely abrogated function of DSSC1 not just SNPs that have a limited impact on expression and function of this gene. My suggestion is to move the analysis of the LOY to a different paper or include this information towards the end of the manuscript as an approach that highlights the relevance of many micronuclei associated genes to human conditions including LOY.

d. Appropriate use of statistics - analyses seemed appropriate

e. conclusions seemed appropriate

f. suggested improvements. I think this is a very large amount of information for such a short format. I have suggested moving most of the LOY analysis to a separate more focused manuscript but if retained, presenting the information towards the end of the manuscript would make the flow better, I think.

g. references seem fine to me

h. clarity- as noted I think the format is too terse and a longer format would probably do a better job of describing the experimental approach and findings.

Author Rebuttals to Initial Comments:**Referee #1 (Remarks to the Author):**

R1.1. "In this study, Adams et al. sought to identify genetic factors that contribute to the formation of micronuclei (MN) in vivo by screening over 6,000 mice across 997 loss-of-function mutant lines. Among these mutant mice, they identified 145 genes that, when knocked out, can lead to either increased or decreased levels of micronucleation in red blood cells. In addition, the authors demonstrated that a subset of these genes overlapped with loci in humans that are susceptible to mosaic loss of Y chromosome (LOY), a biomarker of genomic instability. Additionally, the authors showed that mutant mice that exhibited increased or decreased micronucleation displayed phenotypes that are associated with a wide range of disorders in different systems/organs. The authors then focused on one of the top hits that increased micronucleation—the gene Dscc1. Consistent with previous studies, the absence of DSCC1 leads to DNA damage, slow cell growth, and senescence. In addition, the authors showed that Dscc1 KO mice displayed some developmental defects similar to human cohesinopathies. By performing a CRISPR-Cas9 screen, the authors identified four suppressors of the DSCC1-deficient phenotype. They then showed that by inhibition of one the suppressors, SIRT1, they could rescue the slow cell growth and DNA damage defects of DSCC1-depleted cells. This rescue effect was correlated with the restoration of acetylation of SMC3. Overall, this is a comprehensive screen of almost 1000 mouse mutants and identifies around 150 genes associated with MN formation."

We thank Reviewer 1 for noting that our screen is comprehensive and for outlining the various analyses we performed to explore the data and characterise *DSCC1*, which we implicate as being a powerful regulator of micronucleation. This project represents an enormous amount of work performed on an unprecedented scale and it took more than 10 years to collect all of the mouse data.

R1.2. "However, although this is an impressive effort, I do not feel that as currently presented this is enough of a conceptual advance for the general audience of Nature for a number of reasons. This is not the first time that RBC micronucleation was used to screen for genome instability mutations (e.g., the Chaos mouse mutants). More importantly, the screen has not identified a new functional class of mutations that cause micronucleation- it is already known that many cell division defects cause micronucleation. The phenotype of micronucleation is induced by many types of genome instability and this screen therefore overlaps with other genome stability screens. This impacts the general utility of this study as a comprehensive resource. The authors do report a few strains that reduce micronucleation, which could be novel. But there are only a few such mutations, the dynamic range of the phenotype is small, no functional theme seems to emerge, and these mutations are not functionally characterized in this paper."

We thank Reviewer 1 for noting that our work represents an impressive effort. We have responded to each of the points made below:

1. The Chaos mouse that Reviewer 1 mentions (above) was identified as part of an ENU mutagenesis study published in 2007 [Shima et al., (PMID 17143284)], a paper that has been cited more than 300 times. Very few of the causal alleles from this screen (<5) have ever been identified because to find ENU-induced mutations in this experiment extensive genetic mapping on a hybrid background was required before sequencing could be performed. In contrast, we used knockout/targeted mouse lines and know exactly which gene is causing the micronucleus phenotype in each mutant line. We also generated significant additional phenotyping data, which Shima et al., did not do, noting that their study was just the Chaos mutant. Unlike Shima et al, each line in our study is

available/cryo-preserved for the community. Therefore, our experiment is a significant advance in precision, quality and scale, with a much higher yield of phenotypic mutants (>30x) and extensive data on each mutant.

2. Reviewer 1 makes the observation that we have identified 74 knockouts/mutants with reduced micronucleation, which are completely novel. In response to their request for functional data on this class of micronucleus regulator, we selected 7 genes (10%) and by using gRNA-CRISPR directed editing, have created KO cell lines. We treated these lines with low dose hydroxyurea to induce micronucleation and show that 4 out of 7 genes (*TATDN3*, *HMX3*, *DUSP7* and *PIAS2*) showed significantly reduced micronucleation compared to controls (new Supplementary Fig.1b). Of note, it is possible that the unchanged genes are also involved in MN formation, but that this biology is not captured *in vitro*. To our knowledge, this is the first time genes involved in decreased MN have been described, which will be of great interest to readers of the paper.
3. As we describe in the manuscript, we identified a wealth of mutants showing an increased number of micronuclei, such as knockouts of the genes *Lsm10*, *Prmt7* and *BC013712/C1Orf38*, which have not previously been described as regulators of genomic stability, and therefore illuminate entirely new biology. This fact illustrates the strength of large-scale (unbiased) genetic screens for the identification of new avenues for biological exploration. To further make this point and to provide an example of where new functional themes have emerged, we note that our screen identified *Lsm10*, a gene involved in snRNP processing, a process not previously linked to genomic stability. Thus, our study does reveal new functional groups of micronucleus regulators. We have clarified this in the revised manuscript.
4. The reviewer comments on the “dynamic range” of the MN assay when referring to mutants with reduced micronucleation. It should be noted that we examined micronucleus levels without, for example, irradiation or another DNA-damaging agent to model spontaneous micronucleation at a young age. The average absolute MN% in WT mice in these conditions is 0.24% +/- 0.06 s.d.; the -MN mutants range from 0.12% to 0.19%, and the +MN mutants from 0.30% to 1.18%. These data show that the assay we use is very sensitive. Importantly, many genes that are contributors to human genetic disease modestly alter the level of genomic instability when assessed, like our screen, at the whole organism level. Indeed, one pre-requisite to study a gene in our experiment was that viable mice needed to be generated since germline disruption of genes that profoundly increase genomic instability would likely result in embryonic lethality – this feature allowed us to study genomic instability and associated phenotypes in a way not possible using other approaches (such as *in vitro* systems).

R1.3. “Finally, the findings on SIRT1 suppressing the micronuclei-promoting mutation in DSCC1, which they link to cohesion acetylation, are interesting and could have therapeutic implications. However, this part of the work is not well-developed (for example, there is no biochemical demonstration that SIRT1 deacetylates cohesion).”

We thank the reviewer for noting that the discovery of the relationship between SIRT1 inhibition and decreased micronucleation is interesting and could have important therapeutic implications. We have now taken this observation significantly further as described below and detailed in the revised manuscript.

1. Firstly, as suggested by the reviewer, we looked to see if SIRT1 directly deacetylates SMC3 by using recombinant SIRT1 protein and show that SMC3 is not a direct SIRT1 target (new Supplementary Fig. 9e). We have also examined SMC3 acetylation in the

DSCC1 constitutive KO cells in the RPE-1 *p53* KO background and see the same thing: in the presence of SIRT1i, there is more ac-SMC3 when normalised to total SMC3, independent of the *DSCC1* expression status of the cells. We note that there is also more total SMC3, likely because now it is stabilised (this is a chromatin fraction). To increase our confidence, we performed similar experiments in HEK293 cells and saw the same SMC3 acetylation defect following siRNA-mediated silencing of *DSCC1* and this was rescued by *SIRT1* KO. We present the data below for the reviewers and are happy to add it to the manuscript if they think it is useful.

In panel a, the figure shows a representative SMC3-IP from RPE-1 cells and quantification of three independent experiments to the right (Ac-SMC3/SMC3). In panel b we show a representative western blot performed in HEK293 WT or *SIRT1* KO cells +/- siDSCC1. If the reviewers or editor believe this data would be important to be added to the manuscript, we would be happy to do so.

- We have also asked if p53 (a known SIRT1 target) is responsible for the rescue and show that although *DSCC1* KO *p53* KO cells are viable, they still have a profound cohesion defect that is partially rescued by SIRT1i (Supplementary Fig. 9f). Of note, we have mouse data where we crossed *Dscclko/p53ko* mice and show that p53 loss cannot increase the frequency of live born *Dsccl* homozygotes (if anything survival is worse; see panel a) and it accentuates tumour formation (**p=0.028; see panel b). We provide these data here in our response to the reviewers comments and not in our revised manuscript because we feel these results are tangential to the overall story.

Because, at this point, our data suggests that SIRT1 affects SMC3-Ac via an indirect mechanism, we decided to perform mass-spectrometry analyses and examine acetylation in chromatin fractionated samples. To determine the specificity of SIRT1i, we first performed these experiments in HEK293 WT and *SIRT1* KO cells, with or

without SIRT1i, and thus show that SIRT1i is a very effective inhibitor (new Supplementary Fig. 10 and Extended data 6). More specifically, we identified substrates and unique acetylation sites that are common to *SIRT1* KO and SIRT1i, and only 6 SIRT1i off-targets with 15 acetylation sites. We next performed the same experiment using SIRT1i in RPE-1 *p53* KO cells comparing WT vs. *Dscc1* KO genotypes with or without SIRT1i (Supplementary Fig. 11). Significantly, these two experiments allowed us to identify the most robust SIRT1 targets on chromatin by comparing two independent cell lines: HEK293 and RPE-1 *p53* KO cells. These analyses presented in new Supplementary Fig. 11a show a collection of proteins that are SIRT1 substrates on chromatin. These analyses show known targets (i.e. KAT7), but also some surprising SIRT1 targets, of which perhaps one of the most interesting in this context is NIPBL (Supplementary Fig. 11b). To our knowledge, this is the first time that it has been shown that SIRT1 can deacetylate NIPBL (we have tried to extend this analysis further but the available NIPBL antibodies do not work reliably; this will require more work that we consider outside the scope of this paper).

3. To determine which are the specific K acetylation changes that occur in *DSCC1* KO cells and that are modified by SIRT1i. We identified proteins that are modified by SIRT1i only in the *DSCC1* KO cells (Extended data 7b and Supplementary Fig. 11c). Although these investigations are extensive and additive, we are mindful that they have become tangential to the overall story and we therefore explore them further in a supplementary discussion, rather than the main text of the paper.

As we hope that the reviewer will appreciate, we have now performed a large and almost exhaustive body of additional work that highlights new factors that may explain the SIRT1 rescue we see following *DSCC1* loss.

R1.4. “Additionally, the current evidence suggests that SIRT1 inhibition suppresses the loss of DSCC1 specifically and there is no evidence that it has a general role in improving genome stability (for example, if it suppresses other mutants identified in the screen). This aspect of the manuscript could there be developed as an independent study.”

We thank the reviewer for the observation and suggestion. We agree that this point should be the subject of an independent study. However, to examine this interesting question we decided to examine if SIRT1 inhibition can rescue the micronucleus phenotype associated with other mutants related to cohesion, that were identified in the screen (*CTCF* – intellectual developmental disorder 21, OMIM: 604167; and *SMC3* – CdLS 3, OMIM: 606062). As controls we used other cohesion-associated genes from our mouse KO list which didn't lead to altered MN formation (*MAU2* – no reported disease, OMIM: 614560; and *HDAC8* – CdLS 5, OMIM: 300269), as well as other cohesion genes that are disrupted in CdLS/other cohesion related disorders (*RAD21* – CdLS 4, OMIM: 606462; *SMC5* – Atelis syndrome 2, OMIM: 609386; and *STAG1* – intellectual disability disorder 47, OMIM: 604358). We knocked these genes out with CRISPR-Cas9 and induced MN with low-dose HU treatment using *DSCC1* and *SIRT1* knockouts as controls. These analyses presented in Supplementary Fig. 8e show that SIRT1i can also partially rescue defects associated with *RAD21*, *CTCF* and *SMC3* loss but has no discernible effect in the contexts of *MAU2*, *HDAC8*, *SMC5* or *STAG1* loss. It is perhaps important to note that while *HDAC8* mutations leads to CdLS in humans, its loss in mice does not lead to increased MN formation (see Extended Data 1, Pos Rank 214). Thus, our data reinforce the idea that SIRT1 should be considered for repurposing in CdLS or other cohesion-related disorders, and micronucleus formation could be a useful biomarker.

Specific comments:

R1.5. “SIRT1’s rescue effect on DSCC1-depleted cells is partial. DSCC1 is a component of the RCFCTF18 complex that is involved in DNA replication fork progression. It seems formally possible that this rescue could be explained by the restoration of this function. Depletion of WAPL and PDS5A have been previously shown to rescue the DNA replication defect of DSCC1 KO cells (Ref 44). The authors should determine if depletion of these factors rescues the defects they observe in DSCC1-deficient cells.”

We thank the reviewer for this important observation. To look at the effect of *WAPL* and *PDS5A* we first examined our CRISPR data to see where these genes were ranked. Of note, the library we used for screening does not contain *WAPL* gRNAs, and *PDS5A* is not significantly enriched or depleted. To further explore these two genes, we used siRNA knockdown and show that siWAPL can rescue the cell viability defect of *DSCC1* mutant cells while in our hands siPDS5A could not (Supplementary Fig.8a). To explore this result further we made use of HAP1 knockout *WAPL* and *PDS5A* cells (new Supplementary Fig. 8b) and quantified MN formation following siDSCC1 transfection. These data presented in new Supplementary Fig. 8c show that *WAPL* KO has a minor non-significant effect on MN formation, while *PDS5A* loss leads to a greater number of MN than caused by *DSCC1* loss. That being said, when we quantified the number of MN of siDSCC1/*WAPL* KO and siDSCC1/*PDS5A* KO, we saw that both *WAPL* and *PDS5A* inactivation can rescue MN formation caused by *DSCC1* loss. Why couldn't *PDS5A* silencing rescue the cell viability? It might be that *PDS5A* is important for functions other than replication-dependent cohesion and those are overshadowing the *DSCC1*-loss-dependent replication linked cohesion defect. We thus think our data are consistent with the work in Ref. 46 (Jallepalli Laboratory), and we have noted this in our revised manuscript and discussed these results in detail in the Supplementary Discussion.

R1.6. “The authors demonstrate a correlation between the restoration of acetylation of SMC3 and SIRT1 inhibition in this study. This could be novel and therapeutically important. However, as mentioned above, this aspect of the manuscript is not developed mechanistically.”

We refer to the response in point R1.3.

R1.7. “Moreover, the four suppressors that reduce the DSCC1 KO phenotype seem to have widely varied cellular functions and the other three (TGFB2, KIF25 and CARS) seem unlikely to directly increase cohesion function. A better understanding of why these other mutations suppress DSCC1-deficient cells would enhance the confidence in the specificity of the suppression results.”

We thank the reviewer for this suggestion. We agree with the reviewer that these suppressors might work through different mechanisms. To increase confidence in these hits, we have now validated them by using siRNA and show that all three, *TGFB2*, *CARS* and *KIF25*, partially rescue the *DSCC1*-dependent cellular viability defect (new Supplementary Fig. 8a). While we could speculate on what would be the mechanism, we believe that extensive work on these suppressors are out of the scope of this manuscript.

R.1.8 “Minor issues:

- *“Fig. 2c, A definition of “ γ H2AX positive” should be provided.”*

We are grateful for Reviewer 1 helping us improve the clarity of our manuscript. More specifically we have previously published on a FACS-based γ H2AX analysis and we cite this reference in the manuscript. This paper details the methods and gates we used in our study.

- *“Fig. 2d, Please add a scale bar to the figure.”*

This is very well spotted, and we thank Reviewer 1 for noticing this. We have updated the figure and included a scale bar as suggested.

- *“Fig. 3e, There are no error bars on the left graph except for the DSCC1 KD cell condition and one point on for the WT condition. Please make this consistent among all conditions.”*

Thank you for noting this. We have updated the figure to ensure consistency and included error bars as suggested. We also included each data point in the AUC bar graph.

- *“Fig. 3f, The Y-axis label is confusing. Does it indicate the percentage of iPS cells containing micronuclei? Also, please state what the individual dots represent on the graph.”*

We are grateful to Reviewer 1 for looking at the figures and thinking about how we best convey their message. As suggested, we have updated the Y axis and also the description in the Figure legend to clarify what the dots represent. Of note, the dots represent microscopy fields of view.

- *“Fig 3h, The contrast between the Western blot signal of SMC3 and the background is low. It is difficult to determine if the SMC3 levels are comparable between the SIRT1+ and SIRT1- conditions at 2 days by eye. We suggest that the authors include a quantification of the ac-SMC3 signal normalized to the SMC3 signal.”*

Thank you for this excellent suggestion. As noted in the manuscript we performed this experiment in biological triplicate and as such we have quantified band intensities and placed this analysis in new Supplementary Fig. 9d.

- *“Supplemental Fig. 5 a-b, Please add a scale bar to your images.”*

Thank you so much for noticing this omission. We have updated the figure as suggested.

- *“Supplemental Fig. 6c, These two images appear to be at different magnifications but are shown with the same scale bars. Please clarify.”*

This is very well spotted. We have updated this figure including updated scale bars.

We thank Reviewer 1 for their very helpful and careful comments, which have led us to significantly extend, and improve our manuscript.

Referee #2 (Remarks to the Author):

R2.1 “Adams et al describe an integrative analysis of blood samples from ~1,000 mouse LOF mutants to identify strains associated with accumulation of extranuclear micronuclei (MN), a hallmark of genomic instability that is itself associated with a number of disease states. They then compare screen data to Biobank genetic associations and, in a separate effort, follow up with in vivo and in vitro characterization of one hit, Dscc1, a gene previously known to be involved in sister chromatid cohesion.

A philosophical issue arising from studies of this sort is how much focus in the manuscript to devote to the large-scale screening and data collection/analysis aspects vs. how much functional and mechanistic characterization to apply to select hits to develop the novel biology and to demonstrate the value of the dataset. In this case a substantial effort to screen ~6,000 mice representing ~1,000 genotypes for phenotypes related to abnormal micronucleus formation is analyzed, yielding a very convincing and well-controlled data set that is represented by the volcano plot in Figure 1a. The mouse screens identify, among other things, a collection of genes associated with ‘cohesinopathies’ – disorders related to sister chromatid cohesion (SCC), and the top hit is Dscc1, which has surprisingly not been previously associated with cohesinopathies.”

We thank Reviewer 2 for noting that our study represents a “very convincing and well-controlled data set”. It has been a truly enormous effort to perform this study, and while we focused on *Dscc1*, each of the 145 mutants/knockout lines we identified from the screen are stories in themselves that we hope will be developed by the community for years to come.

It is possible that the reason why *DSCC1* has not been associated with cohesinopathies previously is that homozygous disruption of the gene in humans is likely, as seen in mice, to result in significantly reduced penetrance because most individuals lacking the gene would be predicted to die *in utero*.

R2.2. “The authors compare these results to Biobank data, specifically for genes related to mosaic loss of Y chromosome (LOY), in an effort to demonstrate the broader relevance of the mouse screen to human genetic syndromes. The effort is commendable but the results are suspect. What is the “overrepresentation fold change” of intersected hits relative to expectation? A borderline P-value (0.033) across all 136 genes in tiers 1-3 is pretty weak (Supp Table 1a), and notable that only collectively do the hits cross the admittedly arbitrary 0.05 significance threshold.”

We thank Reviewer 2 for this comment which was also made by Reviewer 4 (R4.3). As suggested, we have moved these data and analysis to the supplementary information (Supplementary Fig. 2) and refined the language used to describe them. As noted by Reviewer 4, loss-of-function mouse mutants might not align with the complex architecture of common genetic variation in the human genome. Alignment with recessive genetic disorders might be higher but this is not captured by the LOY dataset which was generated on a population-ascertained cohort. That said our analysis does suggest some overlap and passes the significance threshold we pre-defined for this analysis.

R2.3. “Given that the rest of the paper addresses the role of DSCC1 gene, this section stands alone with fairly unconvincing associations, and could safely be relegated to supplementary information without loss of impact. The enrichment for GWAS LOY seems well represented in the Circos plot in Fig 1b but the other data sets are unconvincing (related question: what is the meaning of the different arc widths per gene in the Circos

plots?). Figure 1d nicely demonstrates how a single cell-level phenotype can present as multiple seemingly independent organism-level phenotypes/disease states.”

Thank you for this thoughtful comment which we greatly appreciate. As detailed above, we have moved the LOY data/analysis/Circos to the supplementary information as suggested by Reviewers 2 & 4 and discuss it only briefly in the main text – we do this because part of the readership of the paper will be human disease/mouse modellers.

The different widths of the arcs relates to the size of the chromosome. Where there are multiple genes on a chromosome the chromosome is divided accordingly. We have made this clear in the revised (supplementary) figure legend which also points readers at the underlying data.

R2.4. “To pursue in vivo validation and characterization of the Dscc1 hit, the authors then create a Dscc1 knockout mouse and demonstrate the developmental and phenotypic defects associated with the genotype, including those consistent with human cohesinopathies as well as (likely GIN-driven) increased tumor formation in older mice.”

We find this spectrum of phenotypes very interesting and note that the *Dscc1* mutant is of relevance to a range of biological areas including developmental biology, human genetic disease, genomic stability and also tumorigenesis.

R2.5. “Finally, a human iPS cell line was engineered to be DSCC1 deficient, through methods that are not easily found, and screened with a CRISPR knockout library to identify genetic modifiers of DSCC1-induced reduced proliferation (in comparison with a WT screen). This is an excellent idea whose phenotypic suppressor hits, including TGFBR2, SIRT1, KIF25, and CARS, offer an intriguing hypothesis about possible rescue of DSCC1-driven genetic disease. However very little information about this screen is included, and whether its overall quality is sufficient to drive confidence in the results is not a conclusion one can derive from the data presented. The type and degree of DSCC1 “knockdown” is not described.”

We thank Reviewer 2 for this very clear and important comment. We have included further information on the screen metrics and in particular a ROC curve comparing the results of the screen to established essential/non-essential genes, a common way of illustrating screen quality (see below and Github: <https://github.com/team113sanger/Large-scale-analysis-of-genes-that-regulate-micronucleus-formation>). Likewise, in the Methods under “Cell Lines” we have added a description of how the *DSCC1* KD iPS cells were generated to complement the information in Supplementary Figure 7 (we also cite a PhD thesis where these methods are discussed in exhaustive detail). Of note, Supplementary Fig. 7 contains a proteomic analysis of the *DSCC1* KD lines showing that *DSCC1* protein was severely depleted/undetectable and in panel (b) RNA expression analysis of *DSCC1* KD lines indicates just 0.08% and 0.04% of wildtype *DSCC1* transcript levels remain in the two *DSCC1* KD clones used in the paper.

ROC Curve of Screens: To draw these curves we used Bagel 2 essential and non-essential genes (see notebook on Github). This analysis reveals high screen quality as evidenced by sensitivity and specificity values which align- or exceeds many of the screen metrics generated for the DepMap and Project Score resources.

sample_label	sample_group	total_reads	mapped_reads	unmapped_reads	n_guides	n_genes	zero_guides
DSCC1 screen A	DSCC1	51963205	46084284	5878921	101090	18226	532
DSCC1 screen B	DSCC1	47107470	41480729	5626741	101090	18226	559
DSCC1 screen C	DSCC1	56710532	49844705	6865827	101090	18226	518
Plasmid	Plasmid	407114154	365151667	41962487	101090	18226	143
WT screen A	WT	45276397	39103705	6172692	101090	18226	524
WT screen B	WT	46019093	39934889	6084204	101090	18226	493
WT screen C	WT	42828627	37214522	5614105	101090	18226	528

Table: Shows summary metrics of the screen as read counts.

R2.6. "The quality of the CRISPR KO screens in the two cell lines is nowhere present (many metrics exist for such quality scores). The purpose of comparative fitness screens is inherently questionable when, e.g, flow based screens for other markers of GIN might be more appropriate."

As above, we thank Reviewer 2 for asking for further details about the metrics of our CRISPR screens and have provided these in the revised manuscript (and Github: <https://github.com/team113sanger/Large-scale-analysis-of-genes-that-regulate-micronucleus-formation>). *DSCC1*-mutant cells show a range of phenotypes that include a proliferation/growth defect as well as genomic instability. In deciding the screening method, we did consider a flow-based screen but elected to perform a proliferation screen because flow-based screens tend to fall short in terms of the coverage of genes analysed (i.e. they are rarely saturating). This is largely because to do such screens enormous numbers of cells need to be flow sorted. Since GIN levels will directly influence cell proliferation and fitness screens are highly sensitive (and can be saturating), we elected to use this method for our study.

Of note, we show in the manuscript that SIRT1 can rescue the GIN phenotype of a range of cell lines (iPS, HEK293, RPE1, HAP1) which further validates our screening rationale. To further reinforce the quality of the screen data we have now also validated *TGFBR2*, *KIF25* and *CARS* (new Supplementary Fig. 8a).

R2. 7. “Nevertheless, the follow-up characterizing SIRT1 inhibition as a suppressor of DSCC1-dependent phenotype holds promise. What is obviously missing is: does SIRT1 inhibition have any effect in vivo? The authors have the mouse models already; to what extent does Sirt1i (assuming the drug works in mice) rescue Dscc1-/- mice? It is almost as if the authors have reserved a Figure 4 for exactly this experiment.”

We thank reviewer 2 for this comment. Due to logistical issues, including the permanent closure of the Sanger Institute mouse facility during the pandemic, we could not perform this work. We communicated with the editor about this suggestion and her feeling was that taking all factors into account, it is beyond the scope of the current paper.

We thank Reviewer 2 for their very clear and helpful comments, which have led to marked improvements in our manuscript.

Referee #3 (Remarks to the Author):

R3.1. "Micronuclei are a type of genome instability event that occurs in pathological states such as cancer. The authors begin with a survey of several hundred mouse lines to discover a set of gene functions that increase and decrease the number of micronuclei. They follow up with genetic and pharmacologic strategies to test hits related to DNA replication and sister chromatid cohesion. Overall the manuscript is a big step toward discovering genes that regulate the formation of micronuclei. The manuscript is well written in a conservative way which I greatly appreciate. The number of experiments is extensive and I do not suggest adding anything new. The authors describe statistics and technical replicates and use them effectively. It is quite curious that so many different mouse phenotypes co-occur with decreased MN AND increased MN."

We are absolutely delighted with this comment. This project really has been a labour of love over a very long time (>10 years), and we are so pleased that the reviewers have noted the value of the work, the scale, and the efforts we have invested to make sure the statistical analysis has been rigorously performed.

R3.2. "One suggestion is that the authors clarify their working model for DSCC1, SMC3 acetylation, and SIRTUIN 1 in the formation of micronuclei. Do they think loss of DSCC1 leads to double strand breaks during DNA replication that fail to be repaired and that leads to chromosome fragments that missegregate? Or do they think the main problem is that whole sister chromatids fail to be accurately segregated, either through premature sister separation or lagging chromosomes, due to failure to biorient on the mitotic spindle? There is precedence in the literature that loss of SMC3 function leads to increased DNA damage following S phase, lagging chromosomes, and micronuclei (Yueh et al., 2021, Development). Haarhuis et al., Current Biology 2013, showed that WAPL depleted cells accumulate DNA damage and MN. The authors could draw on the existing literature regarding DNA replication, cohesin and MN to formulate a working model."

We thank the reviewer for this point. We took this onboard and based on our data propose a model that is detailed in the Supplementary Discussion.

R3.3. "One minor comment-the authors state that MN contain acentric chromosome fragments (page 3). However, the DNA in the MN does sometimes have centromeres, so they should consider rephrasing."

We thank Reviewer 3 for their excellent comments, which have allowed us to greatly improve our manuscript.

Referee #4 (Remarks to the Author):

R4.1 “Spectacular study of micronuclei in a very large panel of mouse lines followed by extensive studies to characterize the impact that the most significant gene has on genomic instability and further studies of genes that can rescue phenotypes associated with loss of this gene.”

We are delighted with this comment. This project has been an enormous amount of work over more than a decade and we are very pleased with the story presented in the manuscript.

R4.2 “Seems highly original and provides some major insights. That said parts of this paper are not well integrated and seem superfluous. Also, it seems to be written as a brief report and given the breadth of analyses a longer format might be better.”

We thank Reviewer 4 for this comment and as detailed below we have re-worked the text to improve the narrative.

R4.3. “b. The quality of the data seems high and there are a very large number of supplementary data files to back up the findings. That said, I found the GWAS of loss of Y to and some of the subsequent studies to be noncontributory. Loss of Y may be due to the same genes that contribute to development of micronuclei but loss of Y can be due to many other mechanisms(<https://www.nature.com/articles/s41576-019-0202-7> [nature.com]). Therefore, I did not understand the relevance of this analysis to the manuscript. Also, looking at Figure 1, the most significant gene involved in micronuclei is DSSC1 as shown in panel A, but panels B and C do not include that gene at all, which indicates a lack of relevance of the LOY analyses to the rest of the paper. I find this component to be distracting. For analysis of DSSC1 performed on the UKBB, I found the description confusing. The animal model for DSSC1 used loss of function variants. I imagine the authors just used SNPs for the DSSC1 analysis in the UKBB and those results would not be very relevant to the mouse phenotypes.”

Reviewer 4 makes a very good point and on reflection we agree. The mice we have generated are knockout (null) alleles while the architecture of the human genome and the variants in the UKBB dataset likely result in a range of effects on gene function. As suggested by Reviewer 2 we have moved these data and the Circos to the supplementary information and provided a shortened and balanced discussion of how the mouse and human data align.

R4.4. “c. A better approach could be to use the recently released exome sequencing data from UKBB or to use imputed data to identify variants that lead to a nonfunctional variant of DSSC1 and evaluate the phenotypes among individuals with severely abrogated function of DSSC1 not just SNPs that have a limited impact on expression and function of this gene. My suggestion is to move the analysis of the LOY to a different paper or include this information towards the end of the manuscript as an approach that highlights the relevance of many micronuclei associated genes to human conditions including LOY.”

Thank you for this very thoughtful comment. As suggested, we have moved the LOY data to the supplementary Information.

R.4.5. “d. Appropriate use of statistics - analyses seemed appropriate”

We are very pleased with this comment. One of the reasons why we like performing large-scale experiments of this kind is that it gives us the opportunity to use rigorous statistical approaches.

R4.6. “e. conclusions seemed appropriate”

Thank you. As noted by Reviewer 3, we have carefully discussed the results of our study and we are pleased that Reviewer 4 is also in agreement that the conclusions are appropriate.

R4.7 “f. suggested improvements. I think this is a very large amount of information for such a short format. I have suggested moving most of the LOY analysis to a separate more focused manuscript but if retained, presenting the information towards the end of the manuscript would make the flow better, I think.”

We agree with this comment and note that Reviewer 2 also suggested these data be moved the supplementary information. We thank Reviewer 4 for this suggestion, which has led us to improve the clarity of our manuscript.

R4.8. “g. references seem fine to me”

Thank you. We have been careful to ensure we cite the excellent work of others in the field.

R4.9. “h. clarity- as noted I think the format is too terse and a longer format would probably do a better job of describing the experimental approach and findings.”

We thank Reviewer 4 for this very helpful comment. As suggested, we have added a more complete description of the methods and experimental approaches and re-worked the discussion to clarify the narrative.

The comments and suggestions provided by Reviewer 4 have greatly contributed to us improving the clarity of our manuscript, and we are very grateful for their input.

Reviewer Reports on the First Revision:

Referees' comments:

Referee #1 (Remarks to the Author):

The authors have done a quite remarkable job of responding to reviewer critiques and I am swayed that the manuscript will be a valuable contribution that should be published without further experimental work in Nature. There is new analysis that reinforces the idea that some of the mutants decrease the frequency of micronuclei, which is surprising. The underlying mechanism for this remains to be studied, but the results reinforce the value of the resource. The authors have also gone far towards clarifying the result that inhibition of the SIRT1 deacetylase compensates for DSCC1 loss, a striking hit from the original screen. There is biochemistry to show that the SIRT1 effect on SMC3ac is indirect (in contrast to the model favored in the initial submission. there is a proteomic analysis that identifies potential SIRT1 targets that might mediate the DSCC1 suppression. Although this is only a first step towards a mechanism, I feel that the results go far enough for a resource paper. As discussed in my previous review, the SIRT1 results have interesting potential clinical applications. Overall, the authors are to be commended for the strong revision and I support publication.

Referee #2 (Remarks to the Author):

The reviewer thanks the authors for the comprehensive responses to all concerns raised, including adding detail about the CRISPR screens and restructuring the flow of the paper to reduce focus on some weaker areas. No further comments.

Referee #3 (Remarks to the Author):

The authors have responded to my comments and I have no further comments except to say that this is a real tour de force!

Referee #4 (Remarks to the Author):

As reviewer 4, I find that the suggestions I made have been followed and I find the paper an important contribution advancing our understanding of the factors influencing micronuclei development and their impact on phenotypes.